



# Impact of North America on the aerosol composition in the North Atlantic free troposphere

M. Isabel García[1,2], Sergio Rodríguez[1,*], Andrés Alastuey[3]

[1]Izaña Atmospheric Research Centre, AEMET, Associated Unit to CSIC "Studies on Atmospheric Pollution", Santa Cruz de Tenerife, 38001, Spain
[2]Department of Chemistry (T.U. Analytical Chemistry), Faculty of Science, University of La Laguna, La Laguna, 38206, Spain
[3] Institute of Environmental Assessment and Water Research, CSIC, Barcelona, 08034, Spain

*Correspondence to*: Sergio Rodríguez (srodriguezg@aemet.es)

**Abstract.** In the AEROATLAN project we study the composition of aerosols collected over ~5 years at Izaña Observatory (located at ~2400 m a.s.l. in Tenerife, the Canary Islands) under the prevailing westerly airflows typical of the North Atlantic free troposphere at subtropical and mid-latitudes. Mass concentrations of sub10-µm aerosols ($PM_{10}$) carried by westerly winds to Izaña, after transatlantic transport, are typically within the range 1.2 and 4.2 µg·m$^{-3}$ ($20^{th}$ and $80^{th}$ percentiles). The main contributors to background levels of aerosols ($PM_{10}$ within the $1^{st}$ - $50^{th}$ percentiles = 0.15 – 2.54 µg·m$^{-3}$) are North American dust (53%), non-sea-salt-$SO_4^=$ (14%) and organic matter (18%). High $PM_{10}$ events ($75^{th}$ - $95^{th}$ percentiles ≈ 4.0 – 9.0 µg·m$^{-3}$) and are prompted by dust (56%), organic matter (24%) and nss-$SO_4^=$ (9%). These aerosol components experience a seasonal evolution explained by (i) their spatial distribution in North America and (ii) the seasonal shift of the North American outflow, which migrates from low latitudes in winter (~32ºN, January–March) to high latitudes in summer (~52ºN, August–September). The westerlies carry maximum loads of nss-sulphate, ammonium and organic matter in spring (March–May), of North American dust from mid-winter to mid-spring (February–May) and of elemental carbon in summer (August–September). Our results suggest that a significant fraction of organic aerosols may be linked to sources other than combustion (e.g. biogenic); further studies are necessary for this topic. The present study evidences how long-term evolution of the aerosol composition in the North Atlantic free troposphere will be influenced by air quality policies and the use of soils (potential dust emitter) in North America.

## 1 Introduction

The export of aerosols from their source areas impacts on air quality (Chin et al., 2007) and also on climate related processes (Ramanathan et al., 2001) in downwind receptor regions. Exposure to aerosols – or particulate matter (PM) – and reactive gases in ambient air pollution is associated with ~3.7 million deaths a year$^{-1}$, mostly due to ischaemic heart disease (~40%), stroke (~40%), chronic obstructive pulmonary disease (~11%), lung cancer (~6%) and acute lower respiratory infections in children (~3%) according to the World Health Organization (WHO, 2014). Aerosols are of special interest as they may have





an influence on direct radiative transfer and cloud properties by altering the radiative effect and on rain. It is estimated that globally, this influence results in mean radiative forcing due to aerosol-radiation and aerosol-cloud interaction of about -0.9 $W \cdot m^{-2}$, with aerosol-radiation contribution of -0.35 $W \cdot m^{-2}$ as a result of net sulphate contributions (-0.4 ), black carbon (+0.4), nitrate (-0.11), dust (-0.1) and organics (-0.12), according to the Intergovernmental Panel on Climate Change (IPCC,

2013; Myhre et al., 2013).

North America is a major source of aerosols and trace gases (Li et al., 2004; Park et al., 2003, 2004). The export of trace gases to the North Atlantic in the so-called North American outflow (Li et al., 2005) is enhanced by mid-latitude cyclones (Dickerson et al., 1995; Merrill and Moody, 1996; Moody et al., 1996). These cyclones frequently form on the lee side of the

Rocky Mountains and propagate eastward, with associated cold fronts southeastward across the eastern United States (US; Whittaker and Horn, 1984; Zishka and Smith, 1980). The cyclones occur every 5 days – on average – in summer (Li et al., 2005), although in spring that frequency may be even higher. Four airstreams are associated with mid-latitude cyclones: the warm conveyor belt ahead of the cold front, the cold conveyor belt, the dry airstream subsiding behind the cold front, and the post cold front boundary layer airstream (Cooper et al., 2002a, 2002b). The northeastward ascending airstream represented

by the warm conveyor belt prompts the upward transport of pollutants from North America to the free troposphere over the North Atlantic (Eckhardt et al., 2004), where it may connect with the westerly circulation at the north of the Azores High (Li et al., 2005) prompting the transatlantic transport of pollutants; this has been documented for relatively long lifetime (LT) trace gases, such as CO (LT~ 60 days) and $O_3$ (LT~25 days) (Honrath et al., 2004; Owen et al., 2006). Convection is also an important mechanism for ventilation of the boundary layer; the convective outflow prompts the upward transport of

pollutants (Dickerson et al., 1987; Talbot et al., 1998), which may remain over North America for several days prompting ozone production and its subsequent export to the North Atlantic (Li et al., 2005). This mechanism is important in the south-eastern United States in summer, as the warm conveyor belt of the mid-latitude cyclones is shifted northward (Li et al., 2005). There are a number of observation-based evidences on the large-scale impact of the CO and $O_3$ pollution events in the North Atlantic linked to North American fires and pollution export (Parrish et al., 1998; Moody et al., 1996; Honrath et al.,

2004; Owen et al., 2006).

Although aerosols have been less studied, some research has found evidence of their export to the Atlantic in the cyclone modulated North American outflow (Li et al., 2005), even if they have a relatively short lifetime (LT~15 days). By ground based and airborne lidar measurements, Ancellet et al. (2016) detected the transatlantic transport of North American biomass

burning aerosols and dust to the Mediterranean. At Pico Observatory in the Azores, free troposphere transport of North American black carbon aerosols linked to boreal fires (Val Martin et al., 2006) and sulphate, nitrate, elemental carbon and organic aerosols (such as biomass burning) has been detected (Dzepina et al., 2015). Modelling studies have also shown interest in intercontinental transport of aerosols (Park et al., 2004; Chin et al., 2007).





Previous studies on transatlantic transport of North American aerosols have reported on events detected in intensive campaigns, typically lasting from weeks to a few months. In this study we have used a complementary approach, based on long-term records. We analysed the long-term aerosol chemistry register of the Izaña Observatory – located at ~2400 m a.s.l. on the island of Tenerife – with the aim of identifying the composition, potential sources and origin of the aerosols

transported by westerly winds across the North Atlantic. To our knowledge, this is the first study addressing the issue.

## 2 Methods

### 2.1 Sampling site

The Izaña Global Atmospheric Watch (GAW) observatory is located on a mountain ridge (~2400 m.a.s.l.) lying almost permanently above the temperature inversion and marine stratocumulus layer typical of the marine boundary layer (MBL)

top in the subtropics. Buoyant upslope winds develop during daylight, with minimum impact on the aerosol mass concentrations (Rodríguez et al., 2009). At night, upslope winds cease and Izaña is basically exposed to the prevailing westerly free troposphere subsiding airflow.

### 2.2 Sampling and chemical composition

This study is based on a long-term record of chemical composition of PM smaller than 10 µm ($PM_{10}$) and 2.5 µm ($PM_{2.5}$)

aerodynamic diameters at Izaña Observatory. A total of 401 $PM_{10}$ and 315 $PM_{2.5}$ samples were collected and chemically analysed from February 2008 to August 2013.

The samples of $PM_{10}$ were collected on quartz micro fibber filters (150 mm diameter) pre-heated at 205ºC for 5 hours; this procedure removes potentially adsorbed volatile carbon. Aerosol sampling was performed at 30 $m^3 \cdot h^{-1}$ flow rate overnight

(22:00 to 06:00 GMT), under the influence of free troposphere airflows. One $PM_x$ sample was collected every 3 days, except in August, when sampling was daily. Concentrations of $PM_x$ were determined by gravimetry following the EN-14907 procedure (except that filter conditioning was performed at 30–35% relative humidity instead of 50%). The manual gravimetric method is considered optimal for PM concentrations > 10 µg·m$^{-3}$ (EN-14907), so uncertainties are higher below this threshold value (details in section S1 of the supplement). Blank weighting room and blank field filters were collected

and weighted as part of the quality assurance / quality control (QA/QC) protocol.

The methods used in the long-term (30-years) aerosol chemical composition record of Izaña Observatory are described in detail in previous articles (Rodríguez et al., 2015). Briefly, in the study period (2008–2013) soluble species were determined by ion chromatography ($SO_4^=$, $NO_3^-$, $Cl^-$; detection limits 0.113, 0.113 and 0.505 µg·m$^{-3}$, respectively) and selective

electrode ($NH_4^+$; detection limit 0.056 µg·m$^{-3}$). Elemental composition was determined by Inductively Coupled Plasma



Atomic Emission Spectrometry (ICP-AES, IRIS Advantage TJA Solutions, THERMO™) and Inductively Coupled Plasma Mass Spectrometry (ICP-MS, X Series II, THERMO™) after acid digestion of the samples. Organic and elemental carbon (detection limits 0.8 and 0.032 $\mu g \cdot m^{-3}$, respectively) were analysed by thermal-optical transmittance (TOT, Sunset Laboratory Inc.™) following the EUSAAR2 protocol (Cavalli et al., 2009). Because quartz microfiber filters may adsorb

volatile carbon very easily due to the high active surface (Chai et al., 2012), the more unstable part of the organic carbon was discarded based on the results of the field blank filters analysis. Sulphate was split into sea salt sulphate (ss-$SO_4^=$) and non-sea salt sulphate (nss-$SO_4^=$) using the empirical ratio of Na and sulphate ($SO_4^=/Na^+ = 0.25$) in seawater (Gravenhorst et al., 1978), which assumes there is no sulfate enrichment due to gas-to-particle conversion of the oxidation of marine $SO_2$ (Bonsang et al., 1980). Organic matter (OM) was determined by using the ratio OM/OC = 1.8 observed in the North

American aerosols collected at Pico Observatory in the Azores (Dzepina et al., 2015). Blank field filters were subject to gravimetry and chemical analysis, and mean values were subtracted from the $PM_x$ samples.

The chemical composition data were used for a mass closure of $PM_x$ (table 1). The undetermined fraction of PM, i.e. the difference between the gravimetrically determined $PM_x$ and the sum of the chemical compounds, increased under low $PM_x$

conditions. This has already been observed in previous studies (Ripoll et al., 2015) and is attributed to inaccuracies of the manual gravimetric method under low PM concentrations ($< 10$ $\mu g \cdot m^{-3}$) and to the relatively higher contribution of water not fully removed during filter conditioning.

### 2.3 Meteorology, back-trajectories and MCAR plots

We analyzed meteorological re-analysis data from the National Centre for Environmental Prediction / National Centre for

Atmospheric Research (NCEP/NCAR) (Kalnay et al., 1996) to study the processes involved in the export and transatlantic transport of aerosols from North America. The analysis includes geopotential heights, winds and omega (vertical wind) at several standard levels (925, 850 and 700 hPa) and precipitation rates.

Three-dimensional 10-day back-trajectories were computed at 00:00 GMT for Izaña using the meteorological input data

from the European Centre for Medium-Range Weather Forecasts (ECMWF) and Lagrangian model FLEXTRA (Stohl et al., 1995; Sthol and Seibert., 1998). These back-trajectories were used as input in a self-developed Matlab script (The Mathworks, Natick, USA) which segregates air masses coming from North America and the North Atlantic, from those from Africa, attending to the latitude and longitude values along the transport path towards Izaña (details in section S2 of the supplement). The frequency of the westerlies and of the Saharan Air Layer at Izaña is shown in Fig. 1C. Samples of $PM_{10}$

and $PM_{2.5}$ were associated with westerlies and SAL according to the back-trajectories (Fig. S2).





We determined the Median Concentrations At Receptor (MCAR) plots for the main $PM_{10}$ chemical component using the method described by Rodríguez et al. (2011). In these MCAR, the typical (median) concentration of each aerosol component recorded at Izaña, when the air mass has passed by each pixel of the study region, is shown. The MCAR plots were calculated with back-trajectories representative of transatlantic transport from North America. Events linked to (i) back-trajectories from North Africa or (ii) associated with Saharan dust re-circulated over the North Atlantic (exported from North Africa westward and then re-circulated eastward, e.g. as described by Ancellet et al., 2016) were removed; to identify the latter type of events we also used the output forecasts of the BSC-DREAM8b model (Pérez et al., 2006; Basart et al., 2012) storages at the Barcelona Supercomputing Centre website (http://www.bsc.es/earth-sciences/mineral-dust-forecast-system/bsc-dream8b-forecast/north-africa-europe-and-middle-ea-1). Similar plots were used to analyse seasonality of the frequency of westerlies reaching Izaña, which is linked to the export of North American pollutants. Monthly Transport Route Frequency (TRF) plots for the period 2008–2013 were calculated in a similar way to the MCAR plots, but instead of the median concentration, the total number of back-trajectories passing by each cell grid is represented.

### 2.4 Complementary data

We used the Global Fire Emissions Database Version 4 including small fires data (GFEDv4.1s; Randerson et al., 2015) to estimate the average burned fraction of each 0.25º x 0.25º grid cell in North America during the study period (2008–2013). The data set was downloaded from the Oak Ridge National Laboratory Distributed Active Archive Center (ORNL DAAC) for biogeochemical dynamics (https://daac.ornl.gov/cgi-bin/dsviewer.pl?ds_id=1293). We also made use of Level 3 UV Aerosol Index (AI) data, from the Ozone Monitor Instrument spectrometer onboard satellite Aura (OMI 2008–2013), to study the spatial and temporal variability of dust in North America during the study period (2008–2013). The data set was downloaded from the Giovanni online data system of the NASA Goddard Earth Sciences Data and Information Services Center (GES DISC; http://disc.sci.gsfc.nasa.gov/).

### 3 Results and discussion

### 3.1 Chemical characterisation

This study focuses on aerosols transported by the westerlies, i.e. the westerly airstream that flows from North America across the North Atlantic at subtropical and mid-latitudes. Previous studies have shown that the Saharan Air Layer (SAL), i.e. the dusty airstream that expands from North Africa to the Americas, is the most important carrier of aerosols in the tropical and subtropical North Atlantic (Prospero and Carlson, 1972); so first we did a brief comparison of the characteristics of the aerosol composition in these two air streams, with the aim of illustrating the huge differences between them. The frequency of the westerlies and of the Saharan Air Layer at Izaña is shown in Fig. 1C; the westerlies occur with high





frequency throughout the year, with a maximum in April–May and a minimum in July–August when Izaña is mostly within the SAL.

Table 1 shows the median chemical composition and mass closure of the $PM_{10}$ and $PM_{2.5}$ aerosols in samples collected at
Izaña under the SAL and the westerlies (Fig. 1A–B). The transport of particulate pollutants in the SAL had already been studied by Rodríguez et al. (2011). The SAL impacts on Izaña in July and August, and is linked to the northern shift of the Harmattan – trade winds (Fig. 1A). The summer SAL occurs 1–5 km a.s.l. off North Africa; it is associated with air from the Mediterranean flowing south-westward to the Sahara resulting in the emissions and export of dust to the Atlantic above the marine boundary layer. The westerlies occur throughout the year and are associated with airstreams from North America
that, in some cases, may have circulated around the Azores High (Fig. 1A).

Concentrations of bulk $PM_{10}$ and $PM_{2.5}$ are ~ 20 and 10 times higher in the SAL than in the westerlies, respectively (table 1). Mass closure of $PM_x$ accounts for a rather low fraction of the gravimetrically determined $PM_x$ concentrations under westerly conditions (~ 50–70% of $PM_x$, table 1), compared to the SAL (70–90% of $PM_x$). This is attributed to the relatively high
inaccuracy of the manual gravimetric method under low $PM_x$ concentrations described above (see details in section S1 of the supplement). Thus, the sum of the main chemical components ($\sum$ in table 1) is probably a better proxy of the actual bulk $PM_x$ concentrations in the westerlies than the gravimetric PM concentrations.

In the SAL, the $PM_{10}$ aerosol population (median of $\sum$ – sum of the main chemical components – ~ 41 $\mu g \cdot m^{-3}$; table 1) is
basically constituted by dust (78%: 36 $\mu g \cdot m^{-3}$) mixed with organic matter (4.5%: 2.1 $\mu g \cdot m^{-3}$), sulphate (3.8%: 1.8 $\mu g \cdot m^{-3}$), nitrate (1.8%: 0.8 $\mu g \cdot m^{-3}$) and ammonium (0.4%: 0.2 $\mu g \cdot m^{-3}$). In contrast, $PM_{10}$ aerosol in the westerlies (median of $\sum$ – sum of the main chemical components – ~ 1.8 $\mu g \cdot m^{-3}$) is predominantly constituted by dust (44.5%: 1.1 $\mu g \cdot m^{-3}$), organic matter (12.4%: 0.32 $\mu g \cdot m^{-3}$), $nss\text{-}SO_4^=$ (9.8%: 0.25 $\mu g \cdot m^{-3}$) and ammonium (2.2%: 0.06 $\mu g \cdot m^{-3}$). Nitrate in the SAL mostly occurs in the coarse range as non-ammonium salt coating dust particles (see details in Rodríguez et al. 2011). In the westerlies
nitrate concentrations tend to be extremely low; in the few observed nitrate events, it tended to occur in the sub-2.5 µm range, attributed to an ammonium salt. In the SAL, about ¼ of non-sea-salt-sulphate ($nss\text{-}SO_4^=$) is present as ammonium sulphate ($as\text{-}SO_4^=$) linked to anthropogenic sulphur emissions, with the remaining ¾ being non-ammonium sulphate ($na\text{-}SO_4^=$) most probably linked to soil emissions of gypsum / anhydrite soil minerals in beds of Saharan dry lakes (Rodríguez et al., 2011).

Aerosols in the SAL and in the westerlies also exhibit differences in terms of size distribution. PM mass mostly occurs in the sub-2.5µm in the westerlies and in the coarse 2.5–10 µm range in the SAL (table 1).

**3.2 North American large-scale meteorology and airstreams**



The meteorological scenarios that prompt pollutant export events from North America are described in previous studies (Merril and Moody, 1996; Moody et al., 1996; Stohl et al., 2002; Li et al., 2005; Owen et al., 2006); here a complementary view is provided. We analysed how large-scale circulations over North America evolve over the year, more specifically how they may influence on the export of aerosols to the Atlantic. The monthly values of key meteorological fields (geopotential

heights, winds and omega at several standard levels, 925, 850 and 700 hPa and precipitation rates, e.g. Fig. 2A) were determined with the National Center for Environmental Prediction/National Center for Atmospheric Research (NCEP/NCAR) reanalysis data (Kalnay et al., 1996). To facilitate interpretation on how the variability in meteorology may influence the export of aerosols we also plotted (i) the latitudinal range of the westerlies over the eastern coast of North America (observed in the monthly NCEP/NCAR wind fields; Fig. 3A), (ii) the spatial distribution of $SO_2$ according to

Fioletov et al. (2016) (Fig. 3B) and of major aerosol components according to Park et al. (2003, 2004) (Fig. 4) and (iii) the monthly mean values of the omega vertical component of wind in selected domains (Fig. 3C and 3D). Finally, in order to link the export of North American pollutants with transatlantic transport, the Transport Route Frequency (TRF) field was determined for each month based on back-trajectories (Fig. 2B). Figure 2 shows examples for illustrative months (January, April, August and November); additional material is presented in the Supplement (Fig. S3).

During January and February, the North Atlantic anticyclone shifts southward, expanding over the Caribbean, resulting in an intense geopotential / pressure gradient (Fig.S2A) and westerly winds over most North America (Fig.2A1). The main stream of the westerlies (which we will refer to as 'westerly jet') flows from Western Canada (~55ºN) to Eastern US entering the North Atlantic at relatively low latitudes (36–38ºN in the 850hPa standard level; Fig. 2A1). We refer to the westerly jet over

the eastern coast of North America as 'North American outflow', whose latitudinal position over the year is plotted in Fig. 3A. The TRF analysis shows that during this period air from Central and Southern US reaches Izaña (Fig. 2B1).

From March to June, the North Atlantic anticyclone progressively intensifies (Fig.S2B) and the western side of its clockwise atmospheric circulation expands from the inner Gulf of Mexico northward to Central and then to Eastern US, resulting in an

airstream that we have called 'the Gulf inflow', which is observed in the wind fields at the 925, 850 and 700hPa levels (Fig. 2A2). The Gulf inflow is observed from March, when the trade winds building up results in an northward inflow across the coast of Texas, which subsequently turns northeastward over Central US (Arkansas to Tennessee–Indiana) and then eastward resulting in a westerly outflow to the Atlantic by the Eastern coast of North America following the clockwise circulation of the North Atlantic high (Fig. 2B2). The analysis of the (vertical) omega component, at the 925, 850 and 700 hPa levels,

shows that upward movement of air occurs in the regions affected by the Gulf inflow of warm and humid air from Texas to Indiana, in such a way that air masses from the continental boundary layer of Central to North-eastern US may be exported in the westerlies to the North Atlantic free troposphere (Fig. S3B). Observe in Fig. 3C how omega decreases to negative values (net upward movements) from March to May in Central US (domain 1 in Fig. 3D) under the influence of the Gulf inflow; the decrease in omega in this season is also observed to the East (domain 2 in Fig. 3D). This is consistent with the





fact that the storm season occurs in this period (March to June) between Central US (Northern Texas–Kansas) to the west of the Appalachians (Tennessee–Kentucky–Indiana), as reported by NOAA (section S4 of the supplement). Along the path of the Gulf inflow there are a number of sources of aerosols and their precursors – including coal-fired power plants (Fioletov et al., 2016) – whose emissions may be lifted to the mid-troposphere during convective processes (Dickerson et al., 1987;

Talbot et al., 1998), and then exported to the North Atlantic free troposphere by the westerly circulation, which in this period tends to occur 35–45ºN (Fig. 3A). The export of pollutants from Eastern US to the Atlantic is enhanced by eastward moving cyclones, whose tracks typically occur south of 40ºN in this season (Cooper et al., 2002a); Fig. 3B shows an illustration of this scenario (Cooper et al., 2002a, 2002b) and the mean $SO_2$ spatial distribution observed by satellite (Fioletov et al., 2016); a dotted blue line shows the typical eastward track of the cyclones typical in March–April, whereas the blue arrows indicate

a simplified scheme of the associated circulation; the satellite detection of $SO_2$ over the ocean off the coast of Virginia to New Jersey (35–40ºN in Fig. 3B) evidences the importance of the export of this aerosol sulphate precursor to the Atlantic. The back-trajectory based TRF analysis shows that air masses from Central US (e.g. domain 1 in Fig. 3D) and Eastern US (e.g. domain 2 in Fig. 3D) are regularly transported to Izaña (Fig. 2B2). This is the season of maximum frequency of westerlies at this observatory (23–27 days·month$^{-1}$; Fig. 1C1) and that has implications for the export of major aerosol

components, whose concentrations are high in North-eastern US (Fig. 4).

In July and August, the North Atlantic anticyclone shifts northward (Fig.S2C) resulting in intense trade winds over the Caribbean; the Gulf inflow continues blowing northward across the great plains up to Canada where it connects with the main westerlies jet, whereas southern winds prevail along the Eastern coast of the US (Fig. 2A3). In September, the trade

winds, the resulting Gulf inflow and southern winds over the Eastern coast weaken. In this season, the westerlies and the resulting North American outflow shift northward (45–55ºN; Fig. 2A3–2B3, Fig.3A); this is consistent with previous studies showing that cyclone tracks and the resulting warm conveyor belts linked to the export of pollutants tend to occur over Canada (Merry and Moody, 1996; Cooper et al., 2002a). This scenario is illustrated in Fig. 3B, where the cyclone track is highlighted by a dotted red line and the circulation by the red arrow. This is consistent with the seasonal evolution of the

omega vertical wind component, which shows the lowest (and negative) values in August and September in Eastern Canada (domain 4 in Fig. 3C and Fig. 3D), indicating upward movements of the North American outflow. At Izaña, the westerlies occur with a minimum frequency in July (12 days) and August (9 days; Fig.1C1), the period in which the observatory is frequently within the easterly Saharan Air Layer (Rodríguez et al., 2011, 2015). Observe how the air masses from inner North America have a lower impact at Izaña compared to at other periods (Fig.2B3).

From October to December, the North Atlantic high shifts southward, expanding over Southeastern US (Fig.S1D); the Gulf inflow weakens and the westerly wind band shifts progressively southward prompting the transport of air from Central US to the North Atlantic (Fig. 2A4–2B4).




An overall analysis indicates marked seasonality in the atmospheric circulations with potential implications for the export and transatlantic transport of major aerosol components. The westerlies (Fig. 2A–2B), including the North American outflow (Fig. 3), occur at lower (subtropical) latitudes in winter than in summer (mid latitudes). This seasonal shift is also associated with the upward transport of air, which is important in Central US to Eastern US in March–May (domains 1 and 2 in Fig. 3C

and 3D) and shifts northward along Eastern North America through spring and summer (domains 2 to 4 in Fig. 3C and 3D), reaching maximum intensity when the North American outflow occurs over Eastern Canada in August and September (domains 4 in Fig. 3D). This affects which source regions of North America impact downwind North Atlantic free troposphere; air masses from Southern US are transported across the Atlantic in winter (Fig. 2B1), from Central US in spring (Fig. 2B2) and from Canada in summertime (Fig. 2B3). This is consistent with the seasonal shift of cyclone tracks,

westerlies and warm conveyor belt described in previous studies (Stohl, 2001; Cooper et al., 2002a). Of special relevance is the spring, when the westerly jet blows over $SO_2$ source regions coupled with upward movements able to transport aerosols emitted near ground to altitude above the boundary layer (Fig. 3B).

### 3.3 Transatlantic transport of North American aerosols

We studied the seasonal variability of the sub-10 μm aerosol components under westerly airflow conditions at Izaña and its

connection to the transatlantic transport from North America. The chemical composition of 126 samples of $PM_{10}$ collected at Izaña (2008–2013) under westerly airflow conditions, whose back-trajectories are plotted in Fig.S1A, were used.

Under westerly airflow conditions, the time series of the aerosol components typically show a low background level and sporadic peak episodes; for example, nss-$SO_4^=$ shows a background of 0.05–0.15 μg·m$^{-3}$ and peak events 0.5–1.5 μg·m$^{-3}$,

organic matter increase from 0.01–0.2 μg·m$^{-3}$ background level to 1–2.5 μg·m$^{-3}$ peak events, whereas elemental carbon has a background < 0.01 μg·m$^{-3}$ and peak events within the range 0.03–0.1 μg·m$^{-3}$. For each aerosol component, we determined the seasonal evolution of the monthly 30[th], 50[th] and 80[th] percentiles as representative of the background levels, central position of concentration distribution and high concentration episodes, respectively. The 30[th] and 80[th] percentiles plots are presented in the supplement (S5). The Median Concentrations at Receptor (MCAR) plots were determined for the study of

the connection of peak events of aerosol components at Izaña with episodes of North American aerosol export.

### 3.3.1 Sulphate

Figure 5A1 shows the MCAR plot for nss-$SO_4^=$, whereas Fig. 5B1 shows the value of the 50[th] percentile (50[th]P) concentration for each month at Izaña. The monthly 50[th]P of nss-$SO_4^=$ shows high levels from March to July (0.28–0.41 μg·m$^{-3}$), with a maximum in March–May (0.33–0.41 μg·m$^{-3}$), and low levels from September to February (0.06–0.24

μg·m$^{3}$). The March to July period can be considered the high-sulphate concentration season, given that both the monthly background (30[th]P) and median (50[th]P) levels are high in this period (Fig. S5A).




The MCAR plot represents the nss-$SO_4^=$ concentration recorded at Izaña (median value, i.e. 50[th]P) when the airflows (tracked by back-trajectories) have passed by each pixel of the study domain (Fig.5A1). Regions with relatively high nss-$SO_4^=$ concentrations (0.3–0.5 µg·m⁻³ in yellow to red scale) compared to the background (< 0.1 µg·m⁻³ in blue) are connected

to potential transport routes. The MCAR plot suggests that there are two preferential transport paths. The first route points to the transport of nss-$SO_4^=$ from the North-East US at ~40ºN; this is consistent with the high $SO_2$ emissions (Fig. 3B) and high nss-$SO_4^=$ concentrations (Fig. 4A) typical of this region (Fig. 3B) associated with coal burning power plants (Mann et al., 2010; Fioletov et al., 2011, 2016) and the North American outflow (Fig. 3). Because the North American outflow occurs over this region (NE–US) during a great part of the year (Fig. 3), this is probably the most important nss-$SO_4^=$ export region

to the Atlantic. The correlated seasonal evolution of omega in this region (domain 2 in Fig. 3C and 3D) and nss-$SO_4^=$ at Izaña (Fig. 5B1) indicates enhanced upward movements of air from March to July enrich the North Atlantic free troposphere in sulphate aerosols. Maximum nss-$SO_4^=$ occurs from March to May (Fig. 5B1), when upward air movements associated with the Gulf inflow (Fig. 3), cyclones (Cooper et al., 2002b) and the occurrence of the North American outflow over this region enhances the export of regional pollutants (Fig. 3A and Fig. 4A). A second transport pathway is associated with

transatlantic transport at higher latitudes (50ºN) and anticyclonic circulation around the Azores High (Fig. 5A1); this route is associated with the occurrence of the North American outflow over Northern US and Canada, from mid-summer (August) to mid-autumn (November; Fig. 3A1). Observe how the drop in the median (50[th]P: from 0.30 to 0.20 µg·m⁻³ Fig. S5A2) and of the background (30[th]P: from 0.26 to 0.16 µg·m⁻³; Fig. S5A3) nss-$SO_4^=$ concentrations from July to August is associated with the northward shift of the main westerly stream and the North American outflow (Fig. 3A1); from August on the westerly jet

occurs at higher latitudes, over Canada, in regions depleted in nss-$SO_4^=$ compared to NE–US (Fig. 4A), as a result less nss-$SO_4^=$ is exported and transported across the Atlantic.

### 3.3.2 Nitrate

The MCAR plot and the monthly 50[th]P of $NO_3^-$ at Izaña are plotted in Fig. 5A2 and 5B2, respectively. Nitrate was present in extremely low concentrations most of the time. Concentrations were < 0.05 µg·m⁻³ in 97 samples, and > 0.1 µg·m⁻³ in only

21 samples; for this reason the 50[th]P was ~zero for most months (Fig. 5B2). High $NO_3^-$ concentration events (80[th]P > 0.1 µg·m⁻³) were mostly recorded in winter and early spring (January–April, Fig.5B2), when high $NO_3^-$ was within the range 0.2–0.8 µg·m⁻³. This is typical behaviour for ammonium nitrate, which mostly forms under low temperature conditions, whereas gaseous nitric acid prevails in warmer environments (Squizzato et al., 2013). The MCAR plot shows transport of $NO_3^-$ at low latitudes 30–35ºN (Fig.5A2), which is consistent with the circulation of the westerlies (e.g. Fig.2A1 and Fig

2B1) and the North American outflow in winter months (e.g. Fig. 3A1), when most of the above-described high $NO_3^-$ events occur (Fig 5B2). The highest concentrations of nitrate in North America occur in the Central North region (Fig.4B); our results suggest that nitrate export events from North America may be associated with NW winds (e.g. as the main stream of



the westerlies in winter, Fig.2A1) over this high nitrate region (Fig.4B) followed by export at 35–30 ºN (Fig. 5B1) under geopotential / pressure systems that should be studied in future research. The MCAR plot also suggests a second transport route similar to that observed for nss-$SO_4^=$, i.e. transatlantic transport at high latitudes (50ºN) and circulation around the Azores High (Fig. 5B1) which is probably associated with the autumn events (Fig. 5B2), when the North American outflow

occurs over Canada (Fig. 3A1). The nitrate concentrations we observe at Izaña are similarly low to those registered by Dzepina et al. (2015) at the Pico free troposphere site in the Azores linked to long range transport from North America; because nitrate may experience negative artefacts during sampling (Schaap et al., 2004; Vecchi et al., 2009), we cannot discard underestimations and believe that further intercomparison with artefact-free real-time nitrate measurements should be carried out.

### 3.3.3 Ammonium

The MCAR plot and the monthly 50[th]P of $NH_4^+$ at Izaña are shown in Fig. 5A3 and 5B3, respectively. Median (50[th]P) and 80[th]P concentrations present a maximum in April–May (Fig. 5B3 and Fig. S5C1), as nss-$SO_4^=$. The MCAR plot for $NH_4^+$ shows two main transport pathways which resemble those of nss-$SO_4^=$; one transport pathways from North-eastern US at ~40 ºN and a second transport route pointing to the occurrence of the North American outflow by Canada and subsequent

transatlantic transport at high latitudes and circulation around the Azores High (Fig. 5A3). Because of the prevalent extremely low levels of nitrate, ammonium is attributed to ammonium-sulphate in most events. The transport (north-eastward export) routes we observe for nss-$SO_4^=$ and $NH_4^+$ in Eastern US (Alabama–Tennessee–Virginia; Fig. 5A1 and 5A3) are similar to those associated with the passage of spring cyclones and front in the region, prompting the export of pollutants to the Atlantic (Cooper et al., 2002b).

### 3.3.4 Elemental Carbon

Figures 6A1 and 6B1 show the MCAR plot and the monthly 50[th]P and 80[th]P of EC, respectively. The analysis of the monthly 50[th]P and 80[th]P values discloses two relevant periods, associated to high (May–September) and low (January–April) EC events. We associated this seasonal variability of the EC transported by the westerlies to Izaña (i) with the spatial distribution of the EC source regions in North America (Fig. 4D) and (ii) with the seasonal shift of the westerlies (Fig.2A1–2A4) and the

North American outflow (Fig. 3). The highest surface concentrations of EC are estimated to occur in what we have called "EC-rich NE–US regions", which include large urban areas placed 40–45 ºN south of the great lakes to the Atlantic coast (e.g. Chicago, Detroit, New Jersey, Philadelphia and New York; Fig. 4D), linked to fossil fuel combustion (mostly diesel exhaust emissions and coal burning) according to Park et al. (2003).

The season of high EC concentrations at Izaña occurs from May to September, when the westerlies shift northward from 40ºN to 55ºN (Fig. 3A) affecting the "EC-rich NE–US regions" (40–45ºN, Chicago to New York, Fig. 4D). This seasonal



shift is associated with a rise of the upward air movements in Eastern US (domains 2 and 3 in Fig. 3C and 3D), including the "EC-rich NE–US regions" (domain 3 in Fig. 3C and 3D), that enhances the export of EC to the North Atlantic free troposphere in the North American outflow. The highest median EC concentrations at Izaña are observed in August and September (~0.03 $\mu g \cdot m^{-3}$; Fig. 6B1 and Fig. S5D1), when the eastward propagating cyclones over Canada prompt the export

of pollutants from these "EC-rich NE–US regions" to the Gulf of Maine and the Atlantic (Merrill and Moody, 1996), in a scenario illustrated in Fig 3B; the lowest values of omega in Eastern Canada (domain 4 in Fig. 3C and 3D) occur in August and September, which indicates a great potential to lift boundary layer air to the North Atlantic free troposphere. This interpretation is consistent with the MCAR plot for EC, which shows a clear transport pathway (at high latitudes, ~50ºN) from Canada and these "EC-rich NE–US regions" to the Gulf of Maine and then to the Atlantic with subsequent circulation

around the Azores High (Fig. 6A1); this EC transport route is similar to the prevalent transport pathway of August and September (Fig. 2B3).

Low EC concentrations at Izaña occur between January and April, when the westerlies and the North American outflow occur at low latitudes (<40ºN; Fig. 3A), to the south of the EC source regions (40–45ºN; Fig. 4D).

Figure 7 shows the mean burnt fraction (%) of each 0.25º x 0.25º grid cell associated with fires, which occur mainly in Southeastern US in January–February (Fig. 7A); it then spreads northward from March on over Central US to Southern Canada (Fig. 7B) and over Canada and NW US in June–September (Fig. 7C) and then shifts southward to US (Fig. 7D). Boreal fires prompt high EC concentrations in Canada (Park et al., 2005) in NW and North-Central US (Washington, Oregon

and Nevada; Park et al., 2003) that can be exported to the Atlantic in the uplifting North American outflow, potentially contributing to the EC records at Izaña in August and September (Fig. 6B1).

### 3.3.5 Organic Matter

Figures 6A2 and 6B2 show the MCAR plot and the monthly 50thP of OM. This aerosol component shows very marked seasonal evolution, with high levels from January to July, and a maximum from March to May (Fig. 6B2, Fig. S5E1–E3).

This is consistent with the MCAR plot (Fig. 6A2), which shows a transport route from Southeastern US to Izaña at low latitudes (30–40ºN), a common circulation of winter and spring (Fig. 2A1–2A2). From August to December, OM concentrations transported by the westerlies to Izaña are low, associated with the occurrence of the westerlies over North America at high latitudes (Fig.2A3–2A4).

The seasonal evolution of OM is very different (~opposite) to that of EC (Fig. 6A2 and 6B2). Air masses transported from Southeastern US to Izaña (January to April) are rich in OM and relatively poor in EC (Fig. 6A2 and 6B2), whereas the air transported from NE US to Izaña (typically from July to September) is poor in OM and rich in EC (Fig. 6A2 and 6B2). This





is consistent with the spatial distribution of these aerosol species in the US (Fig. 4D and 4E) and suggests that in Southeastern US there are sources of OM that are not, on the other hand, important in EC, and are most probably biogenic emissions. Globally, biogenic volatile organic carbon emissions (BVOCs), i.e. precursors of secondary OA, are comprised of isoprene (~50%), methanol, ethanol, acetaldehyde, acetone, α-pinene, β-pinene, t-β-ocimene, limonene, ethene, and propene

(~30%), and other compounds (mostly terpenoids; ~17%) (Guenther et al., 2012). Biogenic emissions are the principle source of OM in the US, followed by three combustion sources that also emit EC (wildfires, fossil-fuel and bio-fuel) (Park et al., 2003); specifically, in South-eastern US, BVOC emissions are mainly isoprene (81%) and monoterpenes (19%) (Goldstein et al., 2008), with biogenic secondary OA predicted to contribute around 10-20% of PM2.5 mass (Liao et al., 2007). A scenario of biogenic emissions higher in SE–US than in NE–US is consistent with global distribution of the

secondary organic aerosols, whose concentrations are usually higher near to the tropics than at mid latitudes (Guenther et al., 2012; Sindelarova et al., 2014). The importance of the spatial variability of the OM and EC sources and the latitudinal shift of the westerlies over Eastern North America is illustrated from July to August, when a drop in OM concentrations (0.85 to 0.20 $\mu g \cdot m^{-3}$; Fig. 6B2) and an increase in EC concentrations (0.005 to 0.03 $\mu g \cdot m^{-3}$; Fig. 6B1) is associated with the northern shift of the North American outflow (43º to 50ºN; Fig. 3A).

### 3.3.6 Mineral dust

Figures 8A and 8B show the MCAR plot and the monthly $50^{th}$P of calcium, aluminium and the associated bulk dust concentrations. These aerosol components exhibit high concentrations from February to May (Fig. 8B1–B3; Fig. S5F–S5H). The MCAR plot shows a pattern of North American dust export at low latitudes (~35ºN, through North Carolina) towards the north-east which, once over the Atlantic, follow the anticyclonic circulation around the Azores High to Izaña (Fig. 8A1–

A3). We attribute these events to dust emissions in a region that expands from SW Texas northward throughout the High Plains, and subsequent dust export to the Atlantic. Figure 8C shows the Major Dust Activity Frequency (MDAF) detected by satellite; dust activity is observed in SW Texas (Chiguagua–Big Bend Desert) in February, through March to May the activity expands northward across the High Plains (western Texas to Nebraska); other dust sources with lower potential to impact on the North Atlantic are also observed in western US (Great Basin, Mojave Desert and Colorado Plateau). The High

Plains are among the major dust sources in North America, and these sources are considered anthropogenic (linked to agriculture), with maximum activity between February and May (Ginoux et al., 2012). Dust emissions and eastward mobilisation is associated with the intense westerly winds linked to eastward moving cyclones, which also prompt the upward transport of dust to several kilometres above ground, according to Novlan et al. (2007); this scenario is illustrated in Fig. 8A3, showing how the associated air mass track is consistent with the dust export and transatlantic transport route to

Izaña observed in our analysis. During these spring events high dust concentrations (100s of $\mu g \cdot m^{-3}$) are lifted to altitudes 6–12 km a.s.l. over Southern US (Talbot et al., 1998). Upward transport of dust is also associated with convective activity in Central US, Colorado and Oklahoma, in May-June (Corr et al., 2016). The correlation we found between the seasonal





evolution of omega in domain 1 (Fig.3C and 3D) and dust at Izaña (Fig.8B3) supports the idea that from February to May westerly winds and the uplifting of air in the High Plains enrich the North American outflow and the westerly jet in the in dust aerosols.

### 3.3.7 Sea salt

Sea salt concentrations at Izaña are extremely low, with monthly $50^{th}$P values 0.07 to 0.22 µg·m$^{-3}$ between December and May (Fig. 9). These low concentrations are typical of free troposphere sites; in fact, sea salt at Izaña (average = 0.25 µg·m$^{-3}$, median 0.16 µg·m$^{-3}$) is about two orders of magnitude lower than in the marine boundary layer of the Canary Islands (average ~ 11 µg·m$^{-3}$, Querol et al., 2004). The extremely low concentrations of this marine aerosol at Izaña supports our interpretations, i.e. the aerosols transported by the westerlies to Izaña are mostly linked to emissions and upward transport in
continental regions of North America, and not over the ocean.

### 3.3.8 Mass closure of aerosols

Fig. 10 shows the contribution of each species to bulk PM$_{10}$ aerosol mass in samples collected in the westerlies at Izaña; data are classified from the highest to the lowest levels. We considered two approaches: including and excluding mineral dust. When mineral dust is not included (Fig. 10A), the most important contributors to bulk PM$_{10}$ are by far nss-SO$_4^=$ and OM. In
the $1^{st}$–$50^{th}$ percentile range for the sum of aerosol components (a proxy of background levels) - 0.05 to 1.0 µg·m$^{-3}$ - the most important contributors are nss-SO$_4^=$ (0.19 µg·m$^{-3}$ on average, accounting for 38% of the sum of chemical species) and OM (0.14 µg·m$^{-3}$, 30%). In the $75^{st}$–$95^{th}$ percentile range (a proxy of high load of aerosol events) - ~2.0 to 3.6 µg·m$^{-3}$ - the most important contributors are OM (1.43 µg·m$^{-3}$, 57%) and then nss-SO$_4^=$ (0.48 µg·m$^{-3}$, 19%). These results are consistent with previous studies that did not include dust, such as Park et al. (2004, 2005), who focused on the composition of aerosols in the
background boundary layer of US, and Dzepina et al. (2015), who studied the aerosols transported from North America to the North Atlantic free troposphere at Pico observatory in the Azores.

When mineral dust is considered, it becomes the most important contributor to sub-10µm aerosol mass (Fig. 10B). In the $1^{st}$–$50^{th}$ percentile range (a proxy of background levels), bulk PM$_{10}$ = 0.15 – 2.54 µg·m$^{-3}$, the most important contributors to bulk aerosol mass are dust (0.78 µg·m$^{-3}$ on average, accounting for 53% of bulk mass), nss-SO$_4^=$ (0.21 µg·m$^{-3}$, 14%), OM (0.27 µg·m$^{-3}$, 18%) and NH$_4^+$ (0.07 µg·m$^{-3}$, 5%). In the $75^{th}$–$95^{th}$ percentile range (a proxy of aerosol events), bulk PM$_{10}$ = 3.9–8.9 µg·m$^{-3}$, the most important contributors to bulk aerosol mass are dust (2.8 µg·m$^{-3}$, 56%) and OM (1.23 µg·m$^{-3}$, 24%) followed by nss-SO$_4^=$ (0.47 µg·m$^{-3}$, 9%) and NH$_4^+$ (0.1 µg·m$^{-3}$, 2%). The lack of previous studies on transatlantic transport of North American dust make the comparison with previous data difficult; it should be highlighted that Ancellet et al. (2016) detected this transatlantic transport in June (out of the seasonal maximum).
Our overall results evidence that dust and organic matter are the most abundant aerosols transported from North America to the North Atlantic free troposphere.



**4 Conclusions**

A ~5-year record of aerosol chemistry at Izaña Observatory (located at ~2400 m.a.s.l. in Tenerife, the Canary Islands) was used to study the transatlantic transport of aerosols. This study shows that North America is a major source of aerosols,
which are transported by the westerly winds across the North Atlantic free troposphere at subtropical and mid-latitudes. The composition of aerosols carried by the westerlies experiences a marked seasonal evolution which is influenced by (i) the spatial distribution of the aerosol sources in North America and (ii) the seasonal variability of the large-scale meteorology in North America. Of special meteorological relevance is the seasonal shift in the westerly jet and the North American outflow, which migrate from low latitudes in winter (~32ºN, January–March) to high latitudes in summer (~52ºN, August–
September). The export of boundary layer air laden in aerosols to the North Atlantic free troposphere is enhanced by the occurrence of cyclones that move eastward with the westerly atmospheric circulation.

We found that the westerlies carry high loads of:
- mineral dust from February to May, associated with dust emissions in a region that expands from SW Texas
(Chiguagua–Big Bend Desert) northward through the High Plains (western Texas to Nebraska), and subsequent dust export to the Atlantic associated with eastward moving cyclones, westerly winds and the North American outflow, which in this period migrate from 35ºN in February to 40ºN in May,
- non sea-salt-sulphate and ammonium from March to May, when cyclones and the associated outflow occur over North-eastern US, where the highest $SO_2$ emissions occur in North America,
- organic matter from February to May, when cyclones and the associated outflow occur over regions of Eastern US rich in organic aerosols according to previous studies,
- elemental carbon in August and September, when cyclones, the westerly jet and the North American outflow occur at high latitudes (50 to 55ºN) favouring the export of boundary layer air from the regions where the highest concentrations of elemental carbon occur in North America according to previous studies (Chicago to New York,
40–45ºN),

The concentrations of sub10-µm aerosol mass ($PM_{10}$) that reach Izaña Observatory after transatlantic transport typically range between 1.2 and 4.23 µg·m$^{-3}$ (20$^{th}$ and 80$^{th}$ percentiles). The most important contributors to background aerosols (when $PM_{10}$ is within the 1$^{st}$–50$^{th}$ percentiles = 0.15–2.54 µg·m$^{-3}$) are North American dust (53%), organic matter (18%) and non
sea-salt-$SO_4^=$ (14%). High $PM_{10}$ events (75$^{th}$ –95$^{th}$ percentiles = 3.9–8.9 µg·m$^{-3}$) are prompted by dust (56%), organic matter (24%) and nss-$SO_4^=$ (9%). Our results suggest that a significant fraction of organic aerosols may be linked to sources other





than combustion (e.g. biogenic) and that North American dust may be linked to anthropogenic dust sources linked to the use of soil.

The overall results indicate that future long-term evolution of the aerosol composition in the North Atlantic free troposphere will be influenced by air quality policies, with implications for the emissions of aerosol precursors, and the use of potential dust emitter soils in North America. The conversion of natural lands to agriculture and pasturage fields have had a number of environmental implications in North America (Nordstrom et al., 2004, Wu et al., 2007). Research has suggested that this change of land use may increase (MNP, 2006, Lawler et al., 2014 ) and this would have implications for dust emission, air quality and dust impacts on the North Atlantic free troposphere".

*Acknowledgements:* This study is part of the project AEROATLAN (CGL2015-66299-P), funded by the Ministry of Economy and Competitiveness of Spain and the European Regional Development Fund (ERDF). The long-term records of the GAW aerosol program are also funded by AEMET. M.I. García acknowledges the grant of the Canarian Agency for Research, Innovation and Information Society (ACIISI) co-funded by the European Social Funds. The authors gratefully
acknowledge the NOAA/ESRL Physical Sciences Division for the provision of the NCAR/NCEP re-analysis, NILU for providing FLEXTRA back-trajectories based on meteorological data provided from ECMWF (European Centre for Medium Range Weather Forecast), the BSC (Barcelona Supercomputing Centre) for providing DREAM8b model, the Oak Ridge National Laboratory (ORNL) Distributed Active Archive Centre (DAAC) - as part of the NASA Earth Observing System Data and Information System (EOSDIS) - for providing the Global Fire Emissions Database, and the GES-DISC Interactive
Online Visualization ANd aNalysis Infrastructure (Giovanni) - as part of the NASA's Goddard Earth Science (GES) Data and Information Service Centre (DISC) - for the OMI AI data set, and the Storm Prediction Centre - as part of the NOAA National Weather service - for providing the Severe Weather Database Files for US tornadoes. We also thank to Juan José Bustos for the calculation of the back-trajectories, Dr. Javier López-Solano for his assistance with the Aerosol Index data processing and Dr. Y.Boose for providing the picture of the Saharan Air Layer conditions. The excellent work performed by
the staff of Izaña Observatory (C. Bayo, C. Hernández, F. de Ory, V. Carreño, R. del Campo and SIELTEC Canarias) is appreciated.

*Competing interests:* The authors declare that they have no conflict of interest.

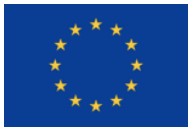
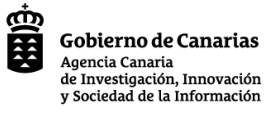
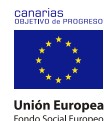

European Regional



Development Fund

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





**Table 1**. Mass closure and median concentrations of $PM_{10}$ and $PM_{2.5}$ components in samples collected when Izaña was (i) within the Saharan Air Layer (SAL) and (ii) within the westerlies (WES). Only days when $PM_{10}$ and $PM_{2.5}$ were sampled simultaneously were taken into account. The percentage of the corresponding parameter with respect to the PM is shown.

| | Saharan Air Layer | | | | westerlies | | | |
| --- | --- | --- | --- | --- | --- | --- | --- | --- |
| | $PM_{10}$ | *%* | $PM_{2.5}$ | *%* | $PM_{10}$ | *%* | $PM_{2.5}$ | *%* |
| NS | 146 | | 146 | | 96 | | 96 | |
| **mass closure** | | | | | | | | |
| PM, µg·m$^{-3}$ | 46.42 | | 21.27 | | 2.54 | | 2.19 | |
| ∑, µg·m$^{-3}$ | 40.87 | *88* | 15.59 | *73* | 1.78 | *70* | 1.05 | *48* |
| undetermined, µg·m$^{-3}$ | 5.55 | *12* | 5.68 | *27* | 0.75 | *30* | 1.14 | *52* |
| **median concentrations** | | | | | | | | |
| dust, µg·m$^{-3}$ | 36.07 | *77.7* | 13.18 | *62.0* | 1.13 | *44.5* | 0.50 | *22.8* |
| sea salt, µg·m$^{-3}$ | < 0.01 | *~0* | < 0.01 | *~0* | 0.01 | *0.5* | 0.01 | *0.5* |
| EC, µg·m$^{-3}$ | < 0.01 | *~0* | < 0.01 | *~0* | 0.02 | *0.6* | 0.01 | *0.3* |
| OM, µg·m$^{-3}$ | 2.07 | *4.5* | 1.00 | *4.7* | 0.32 | *12.4* | 0.26 | *11.7* |
| $NH_4^+$, µg·m$^{-3}$ | 0.18 | *0.4* | 0.17 | *0.8* | 0.06 | *2.2* | 0.06 | *2.7* |
| $NO_3^-$, µg·m$^{-3}$ | 0.82 | *1.8* | 0.17 | *0.8* | < 0.01 | *~0* | < 0.01 | *~0* |
| $SO_4^=$, µg·m$^{-3}$ | 1.73 | *3.8* | 1.06 | *5.0* | 0.25 | *10.0* | 0.22 | *10.0* |
| **sulphate speciation** | | | | | | | | |
| ss-$SO_4^=$, µg·m$^{-3}$ | < 0.01 | *0* | < 0.01 | *0* | < 0.01 | *0* | < 0.01 | *0.1* |
| nss-$SO_4^=$, µg·m$^{-3}$ | 1.72 | *3.7* | 1.05 | *4.9* | 0.25 | *9.8* | 0.22 | *10.0* |
| a-$SO_4^=$, µg·m$^{-3}$ | 0.48 | *1.0* | 0.45 | *2.2* | nd | *nd* | nd | *nd* |
| na-$SO_4^=$, µg·m$^{-3}$ | 1.22 | *2.6* | 0.55 | *2.6* | nd | *nd* | nd | *nd* |

**NS**: number of samples. **PM**: Particulate matter obtained with the gravimetric method. ∑: summation of the major chemical species (dust + sea salt + EC + OM + $NH_4^+$ + $NO_3^-$ + $SO_4^=$). **nd:** not determined.



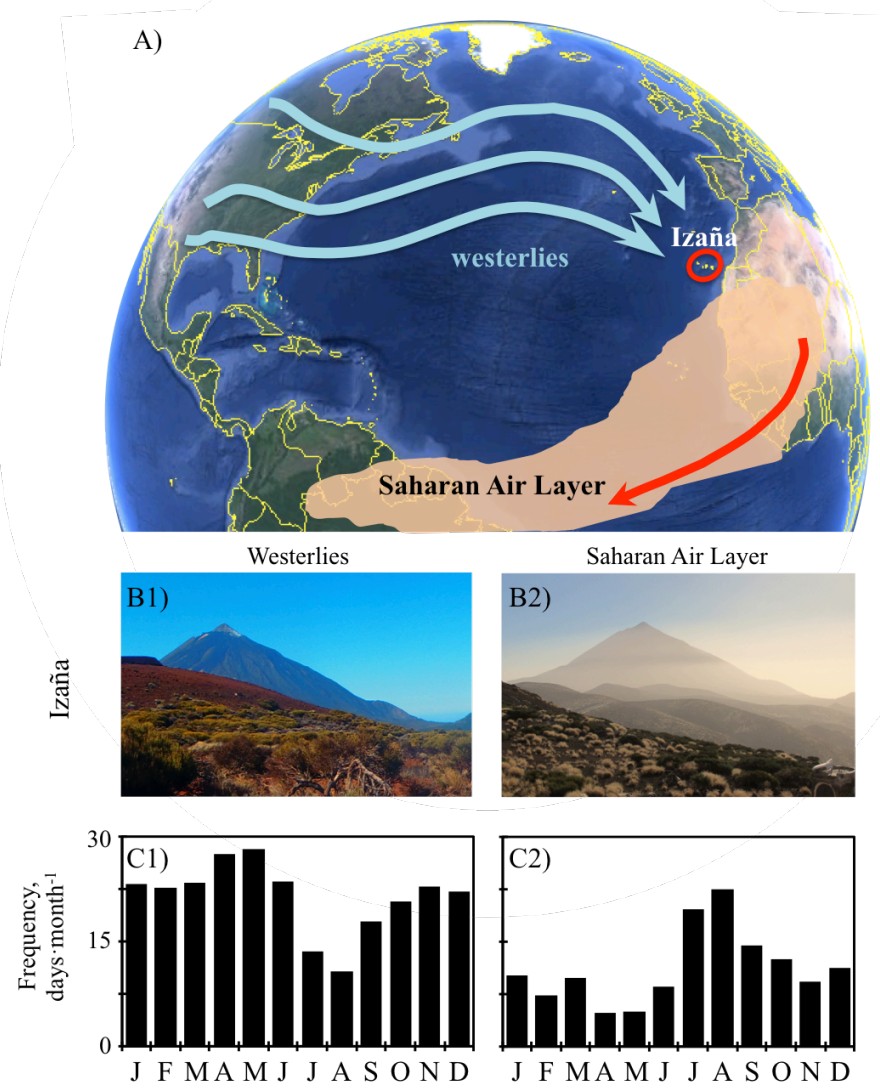

**Figure 1.** (**A**) Location of the Izaña Observatory with an illustration of the Saharan Air Layer and the westerlies. (**B**) View from the Izaña Observatory to the west under the westerly and Saharan Air Layer conditions. (**C**) Monthly frequency (number of days per month) of westerly and Saharan Air Layer at Izaña based on backtrajectories.



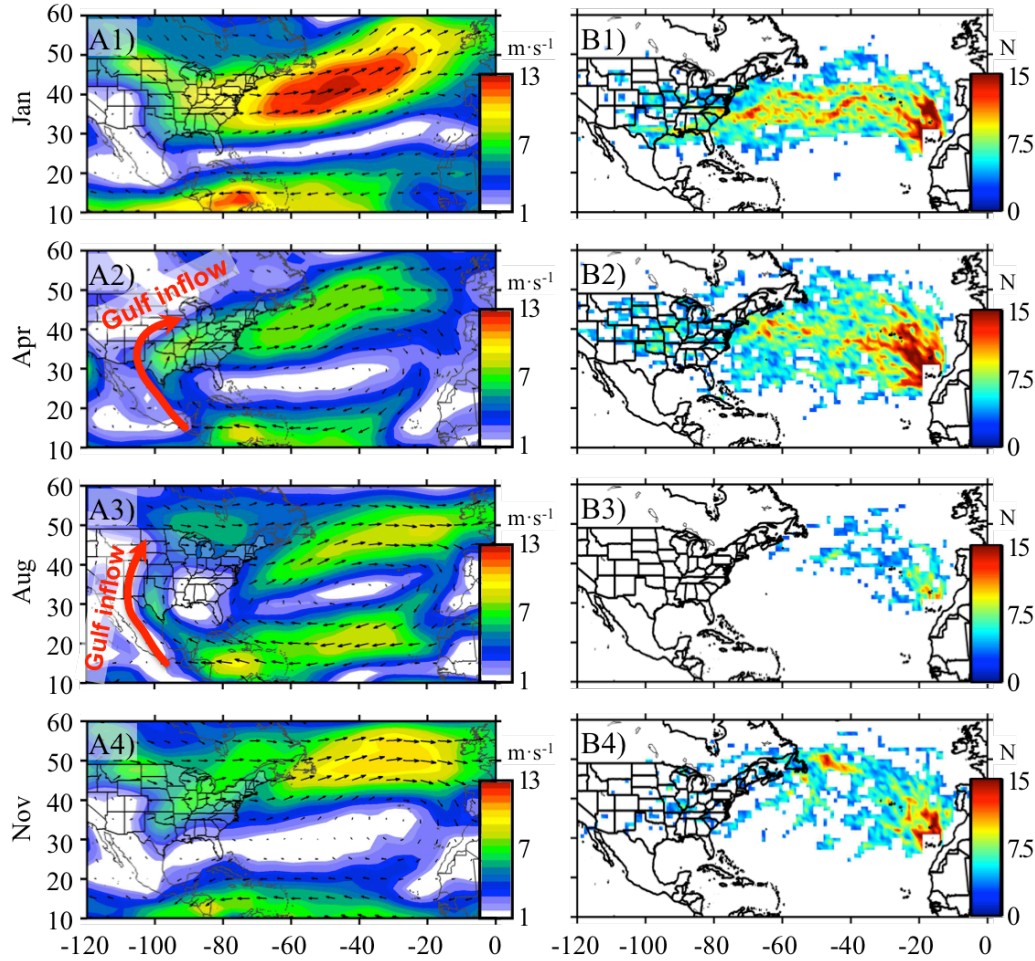

**Figure 2.** (**A**) Wind vector at 850 mb and (**B**) Transport Route Frequency (TRF) for January (Jan), April (Apr), August (Aug) and November (Nov) of the period 2008–2013. The Gulf inflow is highlighted.





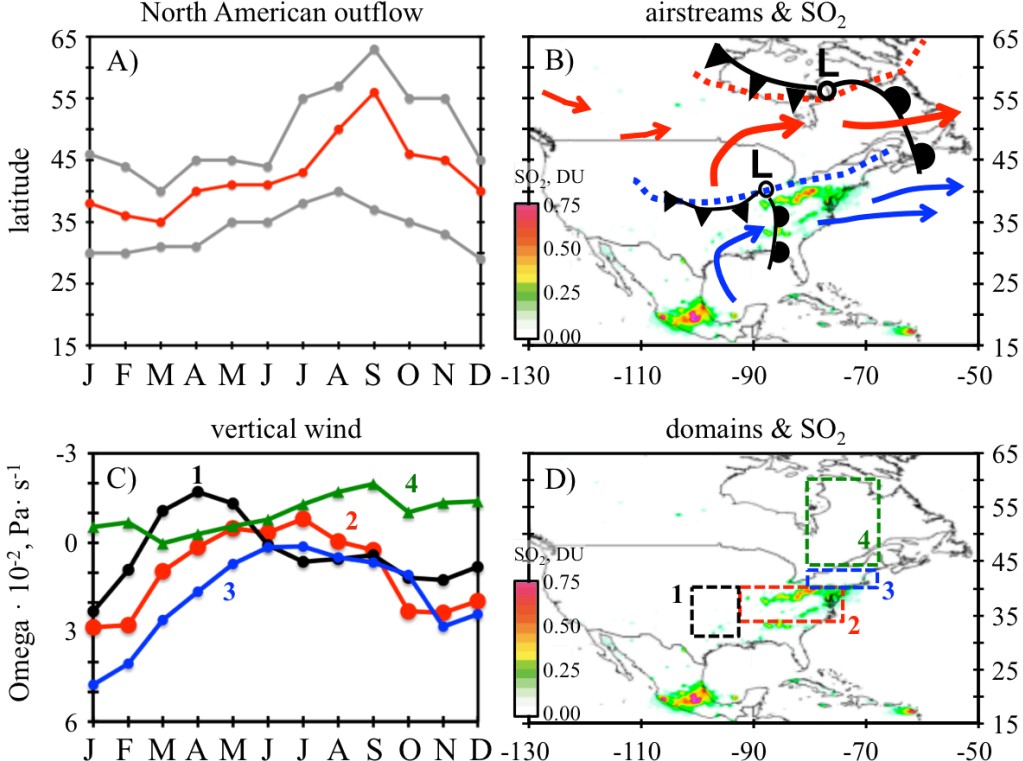

**Figure 3. (A)** Latitudinal ranges at which the westerlies occurs over the Eastern coast of North America. Grey circles: maximum and minimum latitude of the outflow. Red circles: centre of the outflow **(B)** Meteorological scenarios associated with export of pollutants (according to Cooper et al., 2002a, 2002b, Merry and Moody, 1996) and circulations (blue: Jan–May, red: Jul–Aug) illustrated over mean $SO_2$ values observed by satellite by Fioletov et al., (2016) -Copyright by Author(s) 2016. CC Attribution 3.0 License-. **(C)** Monthly average values of the omega vertical wind component at the 850hPa level (negative values indicate upward movements) illustrated in plot (D). **(D)** Domains 1 (32–40º N, 90–100º W), 2 (35–40º N, 75–90º W), 3 (40–43º N, 70–80º W) and 4 (46–60º N, 70–80º W).



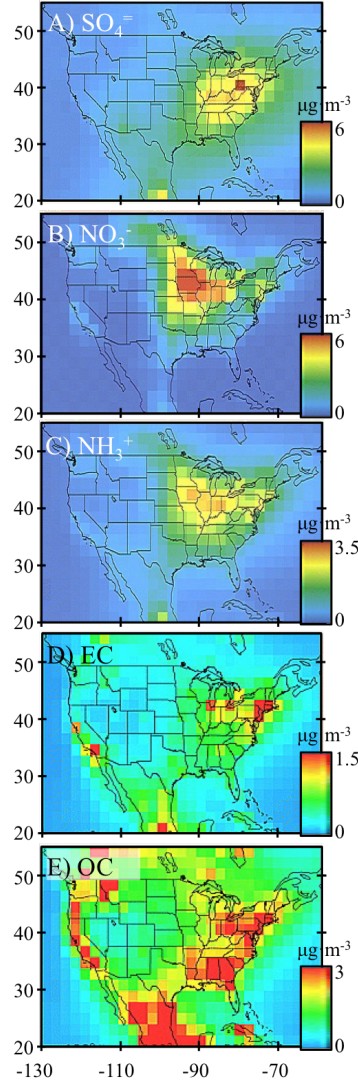

**Figure 4.** Mean surface concentrations of the **(A)** $SO_4^=$, **(B)** $NO_3^-$, **(C)** $NH_3^+$, **(D)** EC and **(E)** OC in North America obtained in previous studies by GEOS-CHEM modelling validated with the observations in the network IMPROVE (Park et al. 2003, 2004, Copyright by the American Geophysical Union).





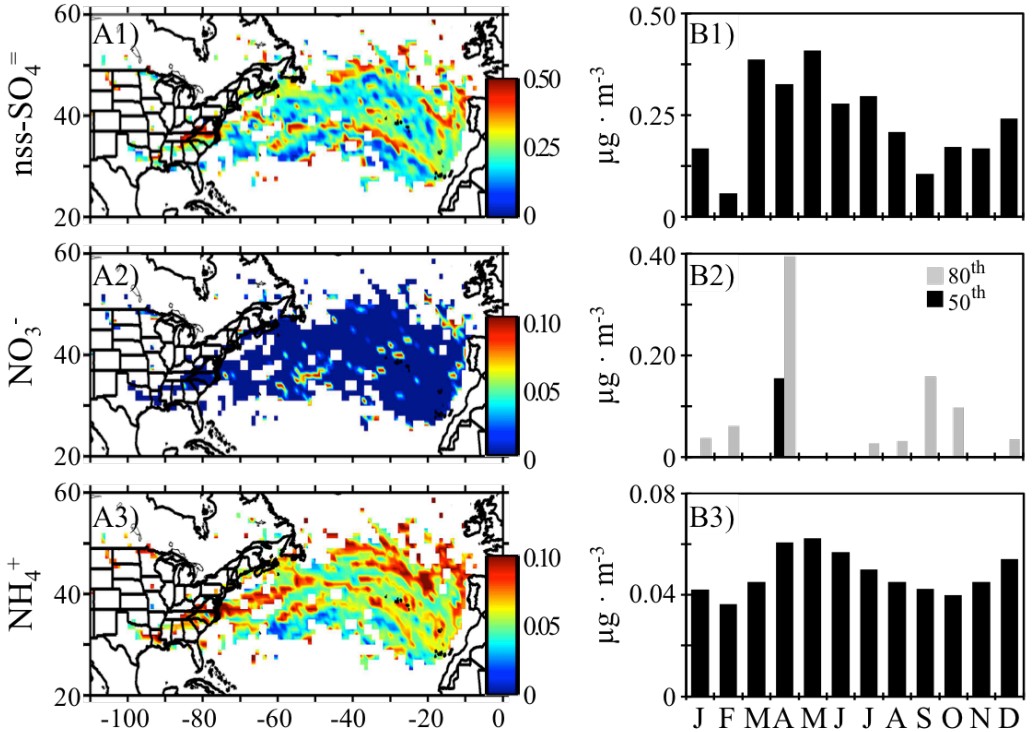

**Figure 5. (A)** Median Concentration At Receptor (MCAR) plots and **(B)** Monthly median distribution for nss-$SO_4^=$, $NO_3^-$, and $NH_4^+$ for the study period. Percentiles 50 and 80 are shown for $NO_3^-$. The MCAR plots maximum concentration tick label includes concentration higher than this upper limit.



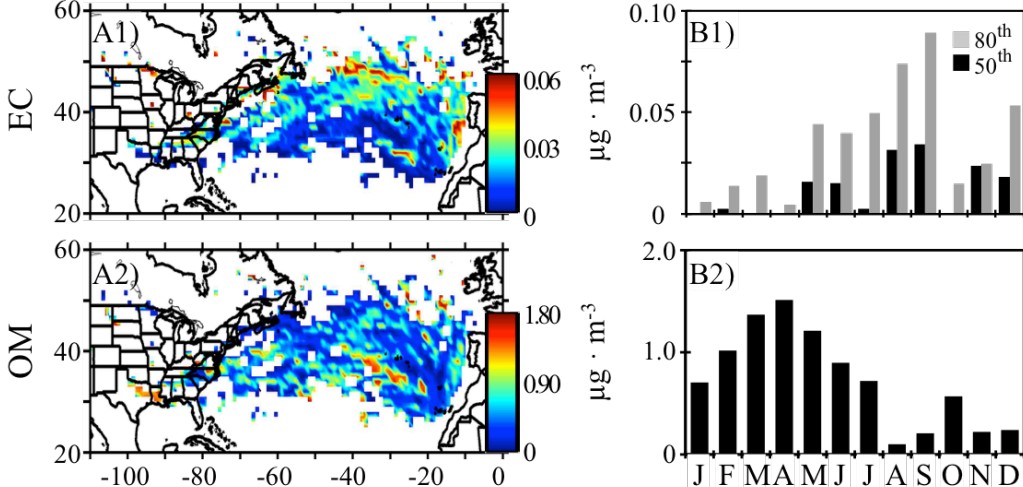

**Figure 6. (A)** Median Concentration At Receptor (MCAR) plots and (**B**) Monthly median distribution for elemental carbon (EC) and organic matter (OM) for the study period. Percentiles 50 and 80 are shown for EC. The MCAR plots maximum concentration tick label includes concentration higher than this upper limit.

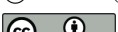



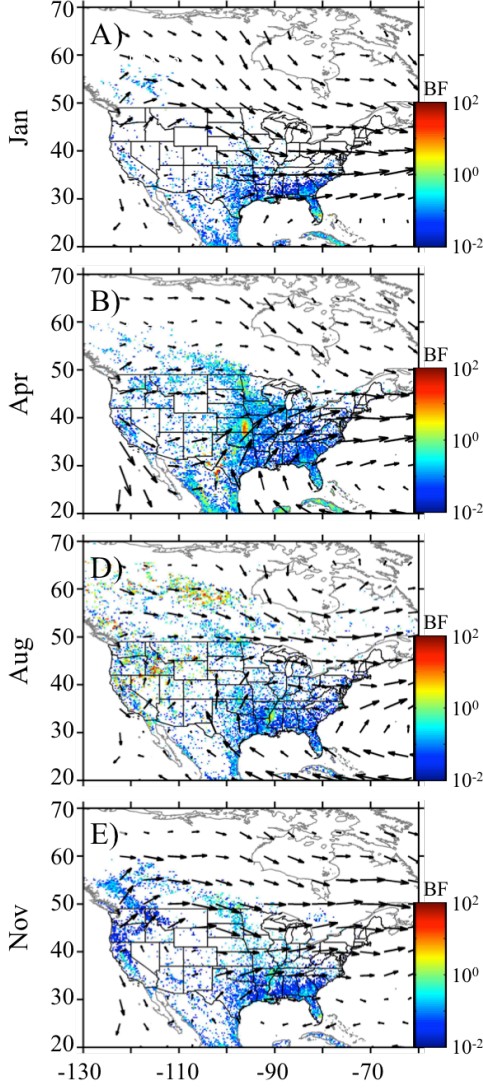

**Figure 7.** Burned fraction (BF) – of each 0.25º x 0.25º grid cell – and vector wind at 850 mb averaged from 2008 to 2013 for (**A**) January, (**B**) April, (**C**) August and (**D**) November. The Global Fire Emissions Database Version 4 including small fires data (GFEDv4.1s; Randerson et al., 2015) was downloaded from the Oak Ridge National Laboratory Distributed Active Archive Centre (ORNL DAAC) for biogeochemical dynamics (https://daac.ornl.gov/cgi-bin/dsviewer.pl?ds_id=1293).



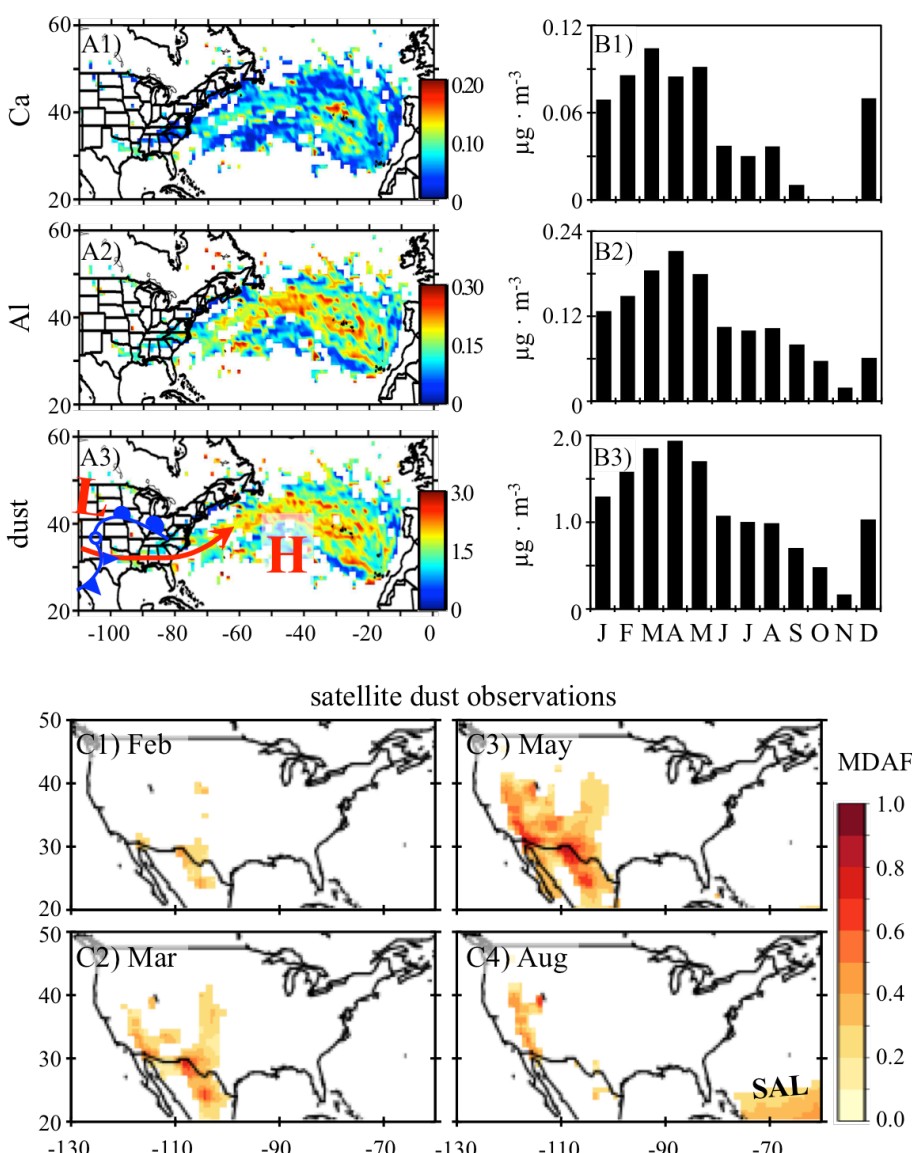

**Figure 8. (A)** Median Concentration At Receptor (MCAR) plots and **(B)** monthly median distribution for calcium (Ca), aluminium (Al) and dust, and **(C)** Mean 2008-2013 Aerosol index averaged (data source: http://disc.sci.gsfc.nasa.gov/). The MCAR plots maximum concentration tick label includes concentration higher than this upper limit.





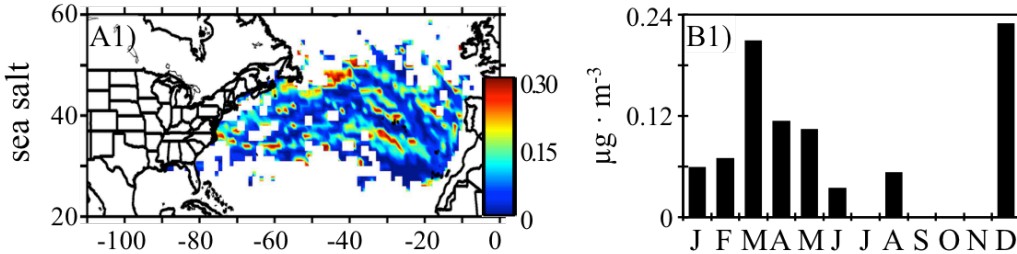

**Figure 9. (A)** Median Concentration At Receptor (MCAR) plots and **(B)** Monthly median distribution for sea salt for the study period. The MCAR plots maximum concentration tick label includes concentration higher than this upper limit.





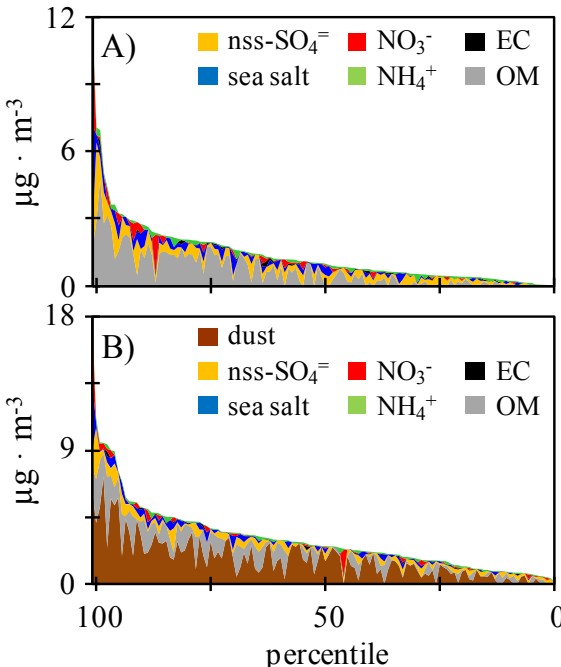

**Figure 10.** Contribution of each aerosol specie to bulk $PM_{10}$ in samples collected at Izaña under westerly airflow conditions; data classified from the highest to the lowest levels. **(A)** Considering mayor components except dust. **(B)** Considering all mayor components including mineral dust.