# Peer review of "Impact of North America on the aerosol composition in the North Atlantic free troposphere"

_Atmospheric Chemistry and Physics, 2017_

## Referee Comment (RC1) · Anonymous Referee #1 · 16 Mar 2017

In this paper, a 5-year record of chemical composition of PM2.5 and PM10 aerosol at Izana Observatory is presented. The composition time series is analyzed in the context of meteorology, calculated back trajectories, and source-receptor plots. The paper shows very clearly that North America is the major source of aerosols sampled at this free tropospheric site. Measured aerosol composition varies seasonally due to the spatial distribution of aerosol sources in North America and seasonally varying large scale meteorology. The paper should be published in ACP once the comments below have been addressed.

Page 2, line 34: Do you mean "evidence of" instead of "interest in"?

Page 3, lines 24 – 25: Change "weighting" to "weighing". Also "weighted" to "weighed".

Page 6, lines 27 – 29: Can the hypothesis that $\frac{3}{4}$ of the non-ammonium sulfate is linked

to soil emissions of gypsum be verified by ratios of calcium or other dust-containing elements to this nss sulfate?

Page 7, line 4: change to "may influence the export".

Figure 2: A symbol indicating the location of the sampling site should be added to each plot – especially Figure 2A1 – A4 – and all similar plots in the paper.

Figure 3 caption: It is not clear how the monthly average values of the omega vertical wind component at the 850 hPa level is "illustrated in plot (D)". Figure 3D appears to be satellite derived SO2 values (based on color bar) and domains.

Page 7, line 17: change to "over most of North America".

Page 8, line 2: A reference should be cited in the main text for the statement "as reported by NOAA".

Page 8, line 9: change to "shows the typical eastward track of the cyclones in March-April"

Page 8, line 13: change to "Spring (March-April) is the season of maximum frequency…."

Page 10, line 20: I'm not sure that "depleted" is the correct word. Change to "…in regions with less nss SO4 compared to NE-US"

Page 10, line 31: State the source of the high nitrate concentrations in the Central north region of North America.

Page 13, lines 1 – 3: Do you mean to say that there are sources of OM that are not related to combustion but, rather, biogenic in origin?

Figure 8 caption: Supply the full name of "MDAF" shown in the color bar for C3. Also what is "Aerosol Index averaged"?

Page 14, lines 5 – 10 and Figure 9: The size range of sea salt discussed and shown

should be provided.

Figure 10 caption: Change "mayor" to "major". Also – it is difficult to see differences in the mass fractions for the non-OM and non-dust components. A logarithmic scale on the y-axis would help.
* * *

---

## Referee Comment (RC2) · Anonymous Referee #2 · 27 Mar 2017

General comment: This manuscript presents the seasonal and spatial aerosol chemical composition in the North Atlantic free troposphere. It comprehensively investigated the main contributions of PM10 and PM2.5 are dust, which accounts for more than 50% of PMx. The seasonal evolution was resulted from the North American outflow, which moved from low latitudes in winter to high latitudes in spring. The author also concluded that the major contributions of organic aerosols are associated with biogenic sources. This manuscript firstly presents the long-term ($\sim$5 years) evolution of aerosol composition in the North Atlantic free troposphere. Overall, this manuscript provides knowledgeable and useful investigation results of aerosol components in North Atlantic free troposphere, which reported the effects of Saharan Air Layer (SAL) and westerlies. The results are clearly presented and the topic is suitable for Atmospheric Chemistry and Physics. The manuscript is recommended for publication after the following minor

comments are addressed.

Minor comments:

1.P3. Line 24: The authors wrote, "the uncertainties may be significant below this threshold vale." What's the percentage of uncertainties in this method? Does the difference of 15% humidity affect the concentrations of PMx?

2.P6. Line 7 : The authors also wrote, " high uncertainty of the manual gravimetric method". How high is the uncertainty?

3.P4. Line 7: The authors used " the empirical ratio of Na and sulfate (=0.25) " to determine the sea salt sulfate and non-sea salt sulfate. Please specify the ratio of Na and sulfate. Is it $SO_4^{2-}$/ $Na^+$ =0.25? Bonsang et al. [1980] reported that the ratios of $SO_4^{2-}$ to $Na^+$ could be greater than 1, and the sulfate contributions could be due to gas-to-particle conversion of the oxidation of marine $SO_2$. It would be good to state the assumptions of this method.

4.P12. Line 27: The OM is rich in southeastern US and could be from biogenic emissions. It would be good to add some examples and references of biogenic emissions.

5.P11. Line1: The author wrote, " because nitrate may experience negative artefacts during sampling we cannot discard underestimations." If there were possible underestimations, would the authors recommend any further methods to improve them?

6.P.15. Line 25: The authors wrote, " the aerosol composition. . .. will be influenced by air quality policies and use of (potential dust emitter) soils in North America." This conclusion points out that the air quality policies and anthropogenic sources are important. I don't find any air quality policies related to dust emission control addressed in this manuscript. The aerosol composition could be influenced by many factors and sources that were addressed in this manuscript. It would be good to emphasize the important contribution.

7.Abstract line 24 : The author wrote, " our results suggest that a significant fraction of

organic aerosols may be linked to sources other than combustion." What's the fraction of organic aerosol do authors suggest?

Technical correction:

1.P13. Line 13: " are Izaña are extremely low" should be "at" Izaña are extremely low.

Reference:

Bonsang, B., B. C. Nguyen, A. Gaudry, and G. Lambert (1980), SUL-FATE ENRICHMENT IN MARINE AEROSOLS OWING TO BIOGENIC GASEOUS SULFUR-COMPOUNDS, Journal of Geophysical Research-Oceans and Atmospheres, 85(NC12), 7410-7416, doi:10.1029/JC085iC12p07410.

---

## Author Comment (AC2) · 21 Apr 2017

reply to comments of reviewer 2 as attached supplement PDF

Please also note the supplement to this comment:
http://www.atmos-chem-phys-discuss.net/acp-2017-60/acp-2017-60-AC2-supplement.pdf

---

## Referee Comment (RC3) · Anonymous Referee #3 · 7 May 2017

The author has shown North America is a major source of north Atlantic free tropospheric aerosol. These aerosol concentrations are enhanced by the propagation of mid-latitude cyclones. Furthermore, there is a seasonality associated with the free tropospheric aerosol composition due to the spatial distribution of aerosol sources in the North Atlantic, and latitudinal shift in the westerly jet from low latitudes in the winter to higher latitudes in the summer. Back trajectories and previous findings on aerosol spatial compositional differences in the United states are linked to explain differences in five years of measured aerosol chemical composition at the Izana Observatory (which is often exposed to free tropospheric air). Seasonal variations in measured free tropospheric aerosol composition correspond to the variation in the north Atlantic outflow Latitude. The manuscript findings are supported by the results and previous literature. I recommend publication after the following minor comments are addressed. Minor

[Figure]

Comments: Line 24 Pg 1- "The present study evidences"- reword Line 28 Pg 1 – remove the word "on" both times it is used in this line. Line 32 Pg 1- reword the last sentence of this page (which continues onto the next page). Line 1-5 Pg 2- separate the second sentence into two sentences. Line 21 Pg 3 – What is EN-14907? Line 3 Pg 4 – shown where? Line 27 Pg 12 – Can you provide references that show evidence of biogenic sources? Figure 10 caption – change "mayor" to "major"

---

## Author Comment (AC3) · 8 May 2017

Answer to comment of **Referee#3**

on "*Impact of North America on the aerosol composition in the North Atlantic free troposphere*" by M.I. García et al.

**Reviewer Comment - OVERVIEW:**

*The author has shown North America is a major source of north Atlantic free tropospheric aerosol. These aerosol concentrations are enhanced by the propagation of mid-latitude cyclones. Furthermore, there is a seasonality associated with the free tropospheric aerosol composition due to the spatial distribution of aerosol sources in the North Atlantic, and latitudinal shift in the westerly jet from low latitudes in the winter to higher latitudes in the summer. Back trajectories and previous findings on aerosol spatial compositional differences in the United States are linked to explain differences in five years of measured aerosol chemical composition at the Izaña Observatory (which is often exposed to free tropospheric air). Seasonal variations in measured free tropospheric aerosol composition correspond to the variation in the north Atlantic outflow Latitude. The manuscript findings are supported by the results and previous literature. I recommend publication after the following minor comments are addressed.*

**REPLY:**

Thanks to Referee #3 for the useful comments that contribute to improve the original manuscript. Please, find below a point-by-point reply to each question and suggestion.

**Minor comments**

1. Line 24 Pg 1- "The present study evidences"- reword

   **REPLY:**
   Thank you very much for your suggestion.

[revised manuscript text omitted]

**7.** Line 27 Pg 12 – Can you provide references that show evidence of biogenic sources?

**REPLY:**
Thank you very much for your suggestion, which was also raised by Reviewer#2. This part of the manuscript needed to be complemented as you point. Also note that in p.13 L.5-8 we stated

previous evidences: "Biogenic emissions are among the principal source of OM in the US, followed by three combustion sources that also emit EC (wildfires, fossil-fuel and bio-fuel) (Park et al., 2003); specifically, in South-eastern US, BVOC emissions are mainly isoprene (81%) and monoterpenes (19%) (Goldstein et al., 2009)."

**CHANGES IN THE MANUSCRIPT** [R3#C7]:
This text was added to the manuscript:
"*This is consistent with previous studies that estimated the contribution of biogenic SOA to OM within the range 50-60% in Southeastern US (Blanchard et al., 2015; Kim et al., 2015; Ying et al., 2015)*".

The following references have been added:

Blanchard, C. L., Hidy, G. M., Shaw, S., Baumann, K., and Edgerton, E. S.: Effects of emission reductions on organic aerosol in the southeastern United States, Atmos. Chem. Phys., 16, 215-238, doi:10.5194/acp-16-215-2016, 2016.

Kim, P. S., Jacob, D. J., Fisher, J. A., Travis, K., Yu, K., Zhu, L., Yantosca, R. M., Sulprizio, M. P., Jimenez, J. L., Campuzano-Jost, P., Froyd, K. D., Liao, J., Hair, J. W., Fenn, M. A., Butler, C. F., Wagner, N. L., Gordon, T. D., Welti, A., Wennberg, P. O., Crounse, J. D., St. Clair, J. M., Teng, A. P., Millet, D. B., Schwarz, J. P., Markovic, M. Z., and Perring, A. E.: Sources, seasonality, and trends of southeast US aerosol: an integrated analysis of surface, aircraft, and satellite observations with the GEOS-Chem chemical transport model, Atmos. Chem. Phys., 15, 10411-10433, doi:10.5194/acp-15-10411-2015, 2015.

Ying, Q., Li, J. and Kota, S. H.: Significant Contributions of Isoprene to Summertime Secondary Organic Aerosol in Eastern United States, Environ. Sci. Technol., 49(13), 7834–7842, doi:10.1021/acs.est.5b02514, 2015.

**8.** Figure 10 caption – change "mayor" to "major"

**REPLY:**
Thank you very much for your observation.

**CHANGES IN THE MANUSCRIPT** [R3#C8]:
"mayor" changed to "major" in Fig. 10 caption.

---

## Author Response (AR1)

Dear Dr. Russell,

I contact you because of the end of the discussion period of the ms <acp-2017-60>. As you can see in the web site, we have already replied to each of the question of the three reviewers. In the reply we have followed the structure (1) comments from Referees, (2) author's response, (3) author's changes in manuscript, recommended by ACP.

We have now prepared a revised version of the manuscript. For preparing this version we have taken into account the comments of the three reviewers. As you can see in the reports, the referees have arisen minor and technical comments and agree publication of this manuscript in ACP. The questions and suggestions of all reviewers have definitively contributed to improve the manuscript.

Please, find attached to this letter (i) the "Authors Response Report", which includes the answers to reviewers 1, 2 and 3, with a reply to each question and the description of the changes in the manuscript as a consequence of each referee comment, and (ii) the "Revised Manuscript and Supplement", where the changes performed (with respect to the ACPD version) are highlighted. We have also followed the structure (1) comments from Referees, (2) author's response, (3) author's changes in manuscript, recommended by ACP, so changes can easily be tracked.

This revised version includes the minor comments and technical changes were introduced to include the comments of the referees.

The latex file of the manuscript and the pdf of the supplement will be uploaded in a further step.

Thanks,
Sergio Rodríguez

**Authors Response Report**

**REPLY TO REVIEWER#1:**

Thanks for the review and the useful comments (listed below) that definitively contribute to improve the original manuscript. Please, find below a point-by-point reply to each question and suggestion.

**1.** Page 2, line 34: Do you mean "evidence of" instead of "interest in"?

**REPLY:**
Yes, thanks.

**CHANGES IN THE MANUSCRIPT** [R1#C1]**:**
"interest in" changed by "*evidence of*".

**2.** Page 3, lines 24 – 25: Change "weighting" to "*weighing*". Also "weighted" to "*weighed*".

**REPLY:**
Thanks for your observation.

**CHANGES IN THE MANUSCRIPT** [R1#C2]**:**
"weighting" changed by "*weighing*" and "weighted" by "*weighed*".

**3.** Page 6, lines 27 – 29: Can the hypothesis that 3/4 of the non-ammonium sulfate is linked to soil emissions of gypsum be verified by ratios of calcium or other dust-containing elements to this nss-sulfate?

**REPLY:**
The scatter plot of Ca versus none-ammonium-sulfate shows a high correlation ($r^2$=0.8) with a slope of about 1.4 (g/g), which is higher than the theoretical one for Ca/$SO_4$ in gypsum (0.4 g/g) due to the presence of Ca with other minerals such as calcite, which is an abundant mineral in Saharan dust (Claquin et al., 1999, Modeling the mineralogy of atmospheric dust sources, Journal of Geophys. Res., 104, 22243-22256); this was discussed in Rodriguez et al. (2011) and more recently in Pérez García-Pando et al. (2016, Predicting the mineral composition of dust aerosols: Insights from elemental composition measured at the Izaña Observatory Authors, Geophys. Res. Lett., 43, no. 19, 10520-10529, doi:10.1002/2016GL069873).

**CHANGES IN THE MANUSCRIPT** [R1#C3]**:**
We have added the reference "*Pérez García-Pando et al. (2016)*" to the main text: "...in beds of Saharan dry lakes (Rodríguez et al., 2011; *Pérez García-Pando et al., 2016)...*"

**4.** Page 7, line 4: change to "may influence the export".

**REPLY:**
Thanks for your observation.

**CHANGES IN THE MANUSCRIPT** [R1#C4]**:**
"may influence on the export" changed by "*may influence the export*".

**5.** Figure 2: A symbol indicating the location of the sampling site should be added to each plot – especially Figure 2A1-A4 – and all similar plots in the paper.

**REPLY:**
Thank you very much for your suggestion, which facilitate the interpretation of the figures.

**CHANGES IN THE MANUSCRIPT** [R1#C5]**:**

The sampling site (Izaña) has been highlighted in Fig.2, Fig.5, Fig.6, Fig.8, Fig.9, and Fig.S3 of the supplement.

**6.** Figure 3 caption: It is not clear how the monthly average values of the omega vertical wind component at the 850hPa level is "illustrated in plot (D)". Figure 3D appears to be satellite derived $SO_2$ values (based on color bar) and domains.

**REPLY:**
Thank you very much for your observation, the caption of that figure in the original version of the manuscript was not enough clear, so we have reworded as follow:

**CHANGES IN THE MANUSCRIPT** [R1#C6]**:**
The text (caption):
"Monthly average values of the omega vertical wind component at the 850hPa level (negative values indicate upward movements) illustrated in plot (D)"
was replaced by:
"Monthly average values of the omega vertical wind component at the 850hPa level (negative values indicate upward movements) *calculated for the domains illustrated* in plot (D)"

**7.** Page 7, line 17: change to "over most of North America".

**REPLY:** Thanks for your observation.

**CHANGES IN THE MANUSCRIPT** [R1#C7]**:**
"over most North America" changed by "over most *of* North America".

**8.** Page 8, line 2: A reference should be cited in the main text for the statement "as reported by NOAA".

**REPLY:**
Thank you very much for your suggestion.

**CHANGES IN THE MANUSCRIPT** [R1#C8]**:**
We have included the reference of the data source "reported by NOAA (*http://www.ncdc.noaa.gov/climate-information/extreme-events/us-tornado-climatology; section S4 of the supplement*)"

**9.** Page 8, line 9: change to "shows the typical eastward track of the cyclones in March-April"

**REPLY:**
Thanks for your observation.

**CHANGES IN THE MANUSCRIPT** [R1#C9]**:**
The text:
"the typical eastward track of the cyclones typical in March-April"
was reworded as:
"the typical eastward track of the cyclones in March-April".

**10.** Page 8, line 13: change to "Spring (March-April) is the season of maximum frequency ..."

**REPLY:** Thanks for your observation.

**CHANGES IN THE MANUSCRIPT** [R1#C10]**:**
"This is the season of maximum frequency of"
changed by
"*Spring (March-April) is the season of maximum frequency*".

**11.** Page 10, line 20: I'm not sure that "depleted" is the correct word. Change to " . . .in regions with less nss-$SO_4$ compared to NE-US"

**REPLY:**
We agree; thanks for your observation.

**CHANGES IN THE MANUSCRIPT** [R1#C11]**:**
"depleted in" changed by "*with less*".

**12.** Page 10, line 31: State the source of the high nitrate concentrations in the Central north region of North America.

**REPLY:**
Thanks for highlighting this issue that will help to clarify this part of the manuscript. There are three factors that contribute to the high concentrations of ammonium nitrate in the Central north region of North America (United States Environmental Protection Agency, 2000, Park et al., 2004): 1) high concentrations of ammonia linked to the livestock and fertilizers, 2) NOx emissions linked to combustion, and 3) thermodynamic conditions favoring the reaction and condensation of ammonia and nitric acid as ammonium nitrate (i.e. enough high gas phase precursor –$NH_3$ and $HNO_3$-, low temperature and enough high relative humidity). This has been cited in the revised version of the manuscript according to this suggestion.

**CHANGES IN THE MANUSCRIPT** [R1#C12]**:**
Text:
"**…**The highest concentrations of nitrate in North America occur in the Central North region (Fig.4B); our results…"
was replaced by:
"**…***High concentrations of nitrate in North America occur in the Central North region (Fig.4B), where conditions favorable for the formation of ammonium nitrate concur (US EPA, 2000; Park et al., 2004): (i) enough high concentrations of gas phase precursors ($NH_3$ linked to emissions in agriculture fields treated with fertilizers and $HNO_3$ due to oxidation of NOx linked to fossil fuel combustion) and (ii) suitable thermodynamic conditions (rather low temperature and enough high relative humidity)…*"

The following reference has been added:
United States Environmental Protection Agency, National air pollutant emissions trends, 1900–1998, EPA-454/R-00-002, Office of Air Qual. Planning and Stand., Research Triangle Park, N. C, 2000.

**13.** Page 13, lines 1-3: Do you mean to say that there are sources of OM that are not related to combustion but, rather, biogenic in origin?

**REPLY:**
Yes, it is what we wanted to say. We have rewritten it in a more simple way.

**CHANGES IN THE MANUSCRIPT** [R1#C13]**:**
The text:
"there are sources of OM that are not, on the other hand, important in EC, and are most probably biogenic emissions"
was reworded as:
"*there is a significant contribution to OM of sources that are not related to combustion, but probably to biogenic emissions*".

**14.** Figure 8 caption: Supply the full name of "MDAF" shown in the color bar for C3. Also what is "Aerosol Index averaged"?

**REPLY:**
Thank you very much for pointing this issue. Fig. 8 caption has been corrected as you point.

**CHANGES IN THE MANUSCRIPT** [R1#C14]**:**

"Mean 2008-2013 Aerosol index averaged" changed by "*Major dust activity frequency (MDAF) for the study period: the number of days with AI values > 1 divided by the total number of days with available AI data in %*".

**15.** Page 14, lines 5-10 and Figure 9: The size range of sea salt discussed and shown should be provided.

**REPLY:**
All results shown in section 3.3 <Transatlatnic transport of North American aerosols> is based on PM10 chemistry, including section "3.3.7 Sea salt" and "Figure 9". It is described in section "2.2 Meteorology, back-trajectories and MCAR plots" and first paragraph of section 3.3.

**CHANGES IN THE MANUSCRIPT** [R1#C15]**:**
Not needed.

**16.** Figure 10 caption: Change "mayor" to "major". Also – it is difficult to see differences in the mass fractions for the non-OM and non-dust components. A logarithmic scale on the y-axis would help.

**REPLY:**
Thanks for your suggestions in which we have spend a time working in. However, this change is not possible in an areal format plot since the log a sum is not the sum of logs. Moreover, that change would smooth the variability of dust and OM, which actually are the most relevant contributors. The fact that the contribution of non-OM and non-dust is rather difficult to see is due to the fact that these species are present in low concentrations (as discussed in the text) and that is the relevant result.

**CHANGES IN THE MANUSCRIPT** [R1#C16]**:**
"mayor" changed by "major".

**REPLY TO REVIEWER#2:**

Thanks to Referee #2 for his useful comments that contribute to improve the original manuscript. Please, find below a point-by-point reply to each question and suggestion.

**Minor comments**

1. P3. Line 24: The authors wrote, "the uncertainties may be significant below this threshold value." What's the percentage of uncertainties in this method? Does the difference of 15% humidity affect the concentrations of PMx?

   **REPLY:**
   For the Izaña sampling conditions (e.g. airflow, duration of the sampling), the uncertainty associated for PM $< 10$ μg·m$^{-3}$ is $\pm 5$ μg·m$^{-3}$. The percentage will depend on the sample mass, ranging from 500% (PM = 1 μg·m$^{-3}$) to 50 % (PM = 10 μg·m$^{-3}$). The standard reference method of Europe set the relative humidity (for filter conditioning and during weighing) to be 45-50%. However, because the sampling at Izaña is usually performed at ambient relative humidity lower than 50%, the long-term aerosol program of Izaña is based on filter conditioning at 30-35% relative humidity. This is already described in section S1 < Uncertainty of the gravimetric method > of the Supplement. The influence of decreasing the conditioning relative humidity on the filter weigh depends on the relative humidity range, such 15% decrease may be relevant when conditioning to rather high relative humidity (50% or higher), but less important when conditioning to low relative humidity (e.g. 20%). The conditioning to 30-35% used in the long-term program of Izaña is suitable for the low relative humidity of the ambient air and is consistent with the measurements of other aerosol properties (optical properties and number size distribution) that are performed after condition the aerosol sample at relative humidity lower than 40% according to GAW standardization.

   **CHANGES IN THE MANUSCRIPT** [R2#C1]:
   This short description has been added in section S1 of the Supplement.

   *"The conditioning to 30-35% used in the long-term program of Izaña is suitable for the low relative humidity of the ambient air and is consistent with the measurements of other aerosol properties (optical properties and number size distribution) that are performed after condition the aerosol sample at relative humidity lower than 40% according to GAW standardization".*

2. P6. Line 7: The authors also wrote, "high uncertainty of the manual gravimetric method". How high is the uncertainty?

   **REPLY:**
   The uncertainty is described in the Supplement. It was determined by using the procedures described in EN 14907. As example, the following table shows the uncertainty associated with the mean concentrations of PM at Izaña shown in Table 1 of the article (see details in the Supplement):

   | | Saharan Air Layer | | Westerlies | |
   |---|---|---|---|---|
   | | PM$_{10}$ | PM$_{2.5}$ | PM$_{10}$ | PM$_{2.5}$ |
   | PM, μg·m$^{-3}$ | $46.42 \pm 4.7$ | $21.27 \pm 4.6$ | $2.54 \pm 4.6$ | $2.19 \pm 4.6$ |

   **CHANGES IN THE MANUSCRIPT** [R2#C2]:
   Not needed.

3. P4. Line 7: The authors used "the empirical ratio of Na and sulfate (=0.25) " to determine the sea salt sulfate and non-sea salt sulfate. Please specify the ratio of Na and sulfate. Is it SO$_4^{2-}$/ Na$^+$ =0.25? Bonsang et al. [1980] reported that the ratios of SO$_4^{2-}$-to Na$^+$ could be greater than 1, and the sulfate contributions could be due to gas-to-particle conversion of the oxidation of marine SO$_2$. It would be good to state the assumptions of this method.

   **REPLY:**
   Thank you very this comment, which is already included in the manuscript (see line 6-9 pag 4).

**CHANGES IN THE MANUSCRIPT** [R2#C3]:
Not needed.

4. P12. Line 27: "The OM is rich in southeastern US and could be from biogenic emissions". It would be good to add some examples and references of biogenic emissions.

**REPLY:**
Thanks for your suggestion. This part of the manuscript actually needed to be complemented as you point.

**CHANGES IN THE MANUSCRIPT** [R2#C4]:
This text was added
"*This is consistent with previous studies that estimated the contribution of biogenic SOA to OM within the range 50-60% in Southeastern US (Blanchard et al., 2015; Kim et al., 2015; Ying et al., 2015)*".

The following references have been added:
Blanchard, C. L., Hidy, G. M., Shaw, S., Baumann, K., and Edgerton, E. S.: Effects of emission reductions on organic aerosol in the southeastern United States, Atmos. Chem. Phys., 16, 215-238, doi:10.5194/acp-16-215-2016, 2016.

Kim, P. S., Jacob, D. J., Fisher, J. A., Travis, K., Yu, K., Zhu, L., Yantosca, R. M., Sulprizio, M. P., Jimenez, J. L., Campuzano-Jost, P., Froyd, K. D., Liao, J., Hair, J. W., Fenn, M. A., Butler, C. F., Wagner, N. L., Gordon, T. D., Welti, A., Wennberg, P. O., Crounse, J. D., St. Clair, J. M., Teng, A. P., Millet, D. B., Schwarz, J. P., Markovic, M. Z., and Perring, A. E.: Sources, seasonality, and trends of southeast US aerosol: an integrated analysis of surface, aircraft, and satellite observations with the GEOS-Chem chemical transport model, Atmos. Chem. Phys., 15, 10411-10433, doi:10.5194/acp-15-10411-2015, 2015.

Ying, Q., Li, J. and Kota, S. H.: Significant Contributions of Isoprene to Summertime Secondary Organic Aerosol in Eastern United States, Environ. Sci. Technol., 49(13), 7834–7842, doi:10.1021/acs.est.5b02514, 2015.

5. P11. Line1: The author wrote, "because nitrate may experience negative artefacts during sampling we cannot discard underestimations." If there were possible underestimations, would the authors recommend any further methods to improve them?

**REPLY:**
Some studies have already shown that quartz filters (as those used in the present study) are the most suitable for avoiding nitrate negative artifacts (e.g. Schaap et al., 2004; Tian et al., 2016). Intercomparison campaigns with real-time measurements in other to evaluate the percentage of losses from the filter-based method could be done in a future.

**REPLY:**
Thank you very much for your observation.

**CHANGES IN THE MANUSCRIPT** [R3#C8]:
"mayor" changed to "major" in Fig. 10 caption.

**Revised Manuscript**

Changes are highlighted in yellow for reviewer#1, in green for reviewer#2, in pink for reviewer#3 and in blue for authors. Brackets indicate the reviewer and comment that prompt the change.

[revised manuscript text omitted]
 [R1#C16 and R3#C8] components except dust. **(B)** Considering all major [R1#C16 and R3#C8] components including mineral dust.

**Revised Supplement**

Changes are highlighted in yellow for reviewer#1, in green for reviewer#2, in pink for reviewer#3 and in blue for authors. Brackets indicate the reviewer and comment that prompt the change.

**S1 Uncertainty of the gravimetric method**

Concentrations of $PM_x$ were determined by gravimetry following the EN-14907 procedure (except that filter conditioning was performed at 30–35% relative humidity instead of 50%). The combined standard uncertainty (uc), associated to a specific $PM_x$ concentration, is expressed as the combination of the individual sources of uncertainty identified in EN-14907; by multiplying uc by the coverage factor k=2, the expanded uncertainty (U) is obtained. U implies that there is a 95% probability that the true value lies within ± U of the measured value and it was calculated for individual samples, with the Izaña sampling conditions (sampling time: 8h; sampler flow: 30 $m^3 \cdot h^{-1}$), as represented in Fig. S1. The expanded uncertainty associated for $PM_x < 10$ $\mu g \cdot m^{-3}$ is ± 5 $\mu g \cdot m^{-3}$. U (%) will depend on the sample mass, with U>50% for $PM_x < 10$ $\mu g \cdot m^{-3}$.

The European standard procedure sets the relative humidity (for filter conditioning and during weighing) to 50% to avoid the effect of water absorption by the filter material, and therefore in the filter mass. The conditioning to 30-35% used in the long-term program of Izaña is suitable for the low relative humidity of the ambient air and is consistent with the measurements of other aerosol properties (optical properties and number size distribution) that are performed after condition the aerosol sample at relative humidity lower than 40% according to GAW standardization [R2#C1].

**S2 Westerlies and Saharan air Layer**

[revised manuscript text omitted]

---

## Author Response (AR2)

Dear Dr. Russell,

Thank you very much for your observations in the manuscript <acp-2017-60>. We have already replied to each of your questions following the structure (1) comments from Editor, (2) author's response, (3) author's changes in manuscript, recommended by ACP.

We have now prepared a revised version of the manuscript. For preparing this version we have taken into account your comments.

Please, find attached to this letter (i) the "Authors Response Report", which includes the answers to Editor, with a reply to each question and the description of the changes in the manuscript as a consequence of each comment, and (ii) the "Revised Manuscript and Supplement", where the changes performed (with respect to the version uploaded after the reviewers comments) are highlighted, so changes can easily be tracked.

This revised version includes the minor comments and changes were introduced to include the comments of the Editor.

The latex file of the manuscript and the pdf of the supplement will be uploaded in a further step.

Thanks,
Sergio Rodríguez

**Authors Response Report**

**REPLY TO EDITOR Dr. Russell**

Thanks to Dr. Russell for her observations.
Please, find below a point-by-point reply to each question and suggestion.

1. Change "artifact free" to online or lower artifact methods, specify.
   **REPLY:**
   Ok, thanks.
   **CHANGES IN THE MANUSCRIPT** [Ed#C1]**:**
   Sentence:
   "further artefact-free real-time nitrate measurements should be included in the long term aerosol measurements program."
   Reworded as:
   "further on-line nitrate measurements (e.g. using an Aerosol Chemical Speciation Monitor) should be included in the long term aerosol measurements program."

2. Fibber >>> fiber.
   **REPLY:**
   Thanks for your observation.
   **CHANGES IN THE MANUSCRIPT** [Ed#C2]**:**
   "fibber" changed by "*fiber*".

3. Artefact >>> artifact.
   **REPLY:**
   Thanks for your observation.
   **CHANGES IN THE MANUSCRIPT** [Ed#C3]**:**
   "artefacts" changed by "*artifacts*".

4. What is evidence for "methanol, ethanol, acetaldehyde, acetone," as precursors of SOA? Guenther et al. say they are bvoc but case for soa precursors is not clear.
   **REPLY:**
   Thanks for your observation. In the manuscript, these compounds are named as BVOCs, but not directly suggested to be SOA precursors: "Globally, biogenic volatile organic carbon emissions (BVOCs), some of which are precursors of secondary OA, are comprised of isoprene (~50%), methanol, ethanol, acetaldehyde, acetone, α-pinene, β-pinene, t-β-ocimene, limonene, ethene, and propene (~30%), and other compounds (mostly terpenoids; ~17%) (Guenther et al., 2012)."
   As we agree that this sentence may lead to confusion, we have reworded it.
   **CHANGES IN THE MANUSCRIPT** [Ed#C4]**:**
   Sentence:
   "Globally, biogenic volatile organic carbon emissions (BVOCs), some of which are precursors of secondary OA, are comprised of isoprene (~50%), methanol, ethanol, acetaldehyde, acetone, α-pinene, β-pinene, t-β-ocimene, limonene, ethene, and propene (~30%), and other compounds (mostly terpenoids; ~17%) (Guenther et al., 2012)."
   Reworded as:
   "Globally, about half of biogenic volatile organic carbon emissions (BVOCs) are SOA precursors, i.e. isoprene (~50%) and α-pinene (~7%) (Guenther et al., 2012)."

**Revised Manuscript**

Changes are highlighted in yellow for Editor Dr. Russell. Brackets indicate the comment that prompt the change.

[revised manuscript text omitted]

**Revised Supplement**

Changes are highlighted in yellow for Editor Dr. Russell. Brackets indicate the comment that prompt the change.

No changes needed.